# A Unified Reasoning Framework for Holistic Zero-Shot Video Anomaly Analysis

**Dongheng Lin**[1,2]    **Mengxue Qu**[1]    **Kunyang Han**[1]
**Jianbo Jiao**[2]    **Xiaojie Jin**[1]    **Yunchao Wei**[1]
[1] Institute of Information Science, Beijing Jiaotong University
[2] The MIx Group, University of Birmingham
{d.lin.2, j.jiao}@bham.ac.uk
{qumengxue, kunyanghan, xiaojie.jin, yunchao.wei}@bjtu.edu.cn

## Abstract

Most video-anomaly research stops at frame-wise detection, offering little insight into why an event is abnormal, typically outputting only frame-wise anomaly scores without spatial or semantic context. Recent video anomaly localization and video anomaly understanding methods improve explainability but remain data-dependent and task-specific. We propose a unified reasoning framework that bridges the gap between temporal detection, spatial localization, and textual explanation. Our approach is built upon a chained test-time reasoning process that sequentially connects these tasks, enabling holistic zero-shot anomaly analysis without any additional training. Specifically, our approach leverages intra-task reasoning to refine temporal detections and inter-task chaining for spatial and semantic understanding, yielding improved interpretability and generalization in a fully zero-shot manner. Without any additional data or gradients, our method achieves state-of-the-art zero-shot performance across multiple video anomaly detection, localization, and explanation benchmarks. The results demonstrate that careful prompt design with task-wise chaining can unlock the reasoning power of foundation models, enabling practical, interpretable video anomaly analysis in a fully zero-shot manner. Project Page: `https://rathgrith.github.io/Unified_Frame_VAA/`.

## 1 Introduction

Video anomaly analysis is a key application of computer vision for public security. Most early works formulate the task as temporal *Video Anomaly Detection* (VAD): mark the segments whose behavior deviates from learned normal patterns. Traditional detectors have reached high performance on benchmarks, yet they output only frame-wise scores and provide no insight into why the segment is abnormal. These limitations of interpretability motivate a broader shift from temporal detection to more downstream anomaly analysis tasks with user-friendly and explainable outputs, including spatial Video Anomaly Localization (VAL) [Liu and Ma, 2019, Weng et al., 2022] and textual Video Anomaly Understanding (VAU) tasks [Du et al., 2024, Tang et al., 2024, Zhang et al., 2024b] utilizing fine-tuned MLLMs. While these works provide either spatial or textual cues for better explainability to video anomalies separately, the previous works were mostly focused on a certain type of downstream tasks, which do not provide a holistic analysis to video anomalies, resulting in *"incompleteness"* from existing video anomaly analysis methods.

A further challenge is the heavy reliance on dataset-specific supervision. Traditional VAD and VAL models require temporal masks or spatial bounding boxes, yet anomaly definitions vary widely across datasets [Wu et al., 2020, Lu et al., 2013, Mahadevan et al., 2010], so a model tuned on one domain

39th Conference on Neural Information Processing Systems (NeurIPS 2025).

Table 1: **Comparison of scopes and requirements of recent VLM-based methods.** ✓ = supported tasks, ✗ = not supported. Our framework is the only strictly zero-shot approach that handles all three.

| Method | Supervision | Fine-tuning | Temporal | Spatial | Textual |
|---|---|---|---|---|---|
| LAVAD [Zanella et al., 2024] | None | None | ✓ | ✗ | ✗ |
| CUVA [Du et al., 2024] | Text | Prompt-tuning | ✗ | ✗ | ✓ |
| STPrompt [Wu et al., 2024b] | Weak class (closed-set) | Prompt-tuning | ✓ | ✓ | ✗ |
| Hawk [Tang et al., 2024] | Instr. tuning | Projection | ✗ | ✗ | ✓ |
| HolmesVAU [Zhang et al., 2024b] | Instr. tuning | LoRA | ✓ | ✗ | ✓ |
| VERA [Ye et al., 2025] | Weak class | Verbalized prompt learning | ✓ | ✗ | ✗ |
| **Ours** | **None** | **None** | ✓ | ✓ | ✓ |

often fails on another [Wu et al., 2024a]. Also, in real-world applications, due to privacy and security concerns, the training data could be unavailable for some sensitive scenes. As partial remedies, recent work has explored zero- and few-shot approaches using frozen vision-language backbones or MLLMs as we summarized in Table 1. We observed that most of the VLM-based works have limited task scope and still rely on annotated datasets. The only strictly zero-shot method is solely focusing on temporal VAD which makes it less user-friendly [Zanella et al., 2024]. For prompt-based methods [Yang et al., 2024, Ye et al., 2025, Wu et al., 2024b], they inevitably require induction on an annotated training set, which comes at the cost of generality as prompts are often learned to be task/domain-specific. This generality problem even exacerbates for instruct-tuned MLLMs [Tang et al., 2024, Zhang et al., 2024b] which are optimized to return answers from seen QA pairs focused on describing a closed set of anomaly types [Ding and Wang, 2024].

In recognition of these problems, given that multimodal LLMs already encode rich visual-semantic priors for commonsense reasoning [Zhao et al., 2023, Zhang et al., 2025b, Ren et al., 2025], fine-tuning may be unnecessary for certain tasks, as long as we can effectively reason about task contexts at test time [Minaee et al., 2025, Ma et al., 2024]. Specifically for video anomaly analysis, we may consider each of the previous benchmark tasks as answering specific questions *(When, where, what, and why?)* about visual anomalies, among which each can be seen as a sub-problem contributing to holistic analysis. Therefore, solving these tasks represents naturally stratified reasoning contexts contributing towards holistic anomaly analysis. Inspired by this, we propose a **unified reasoning-driven chain framework** that conditionally connects different MLLM-based task solvers during test time.

Specifically, our framework operates systematically across three clearly defined stages rather than merely concatenating separate tasks. First, an initial Video Anomaly Detection (VAD) computes a surrogate anomaly probability at the video level and extracts a contextual tag list corresponding to the most suspicious segments, thereby providing individualized context cues for each sample. Following this, a score-gated refinement utilizes both the contextual tag list and preliminary anomaly scores to perform conditional score adjustments, refining the VAD task based on the inferred contexts. Lastly, the final anomaly scores and contextual tag lists jointly guide the downstream spatial Video Anomaly Localization (VAL) and further textual Video Anomaly Understanding (VAU) tasks, where textual and visual prompts are dynamically refined based on the VAD scores. In summary, each stage of our framework employs frozen Vision-Language Models (VLMs), with dynamic prompts iteratively inferred from preceding stages.

We conduct extensive experiments on UCF-Crime, XD-Violence, UBnormal and MSAD [Sultani et al., 2018, Wu et al., 2020, Acsintoae et al., 2022, Zhu et al., 2024]. The proposed framework achieves state-of-the-art performance on three separate tasks under a zero-shot setting, achieving an overall 4-6% AUC improvement on VAD, and consistent improvements over diverse metrics for VAL and VAU tasks. These results show that our training-free, unified video anomaly analysis framework is interpretable, extensible, and robust across various domains and tasks.

## 2 Related Works

**Traditional video anomaly analysis.** Early Video Anomaly Detection (VAD) works typically fall into three major supervision regimes: *one-class* models trained only on normal clips and used compact embeddings or memory banks to detect outliers [Sohrab et al., 2018, Wang and Cherian, 2019, Micorek et al., 2024]; fully *unsupervised* methods rely on reconstruction or future-frame

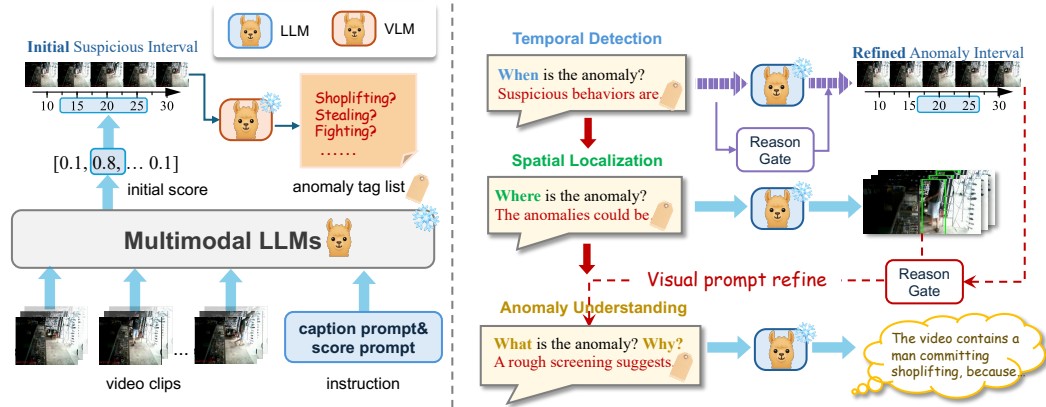

Figure 1: **Overview of the unified holistic anomaly analysis framework. Left:** A preliminary step extracting the most suspicious intervals of a video and extracts anomaly tag lists reflecting possible anomaly contexts. **Right:** Illustration of how the priors are used to refine each of the tasks. Low-confidence samples in Temporal VAD are refined by a selective Intra-Task Reasoning step. The Inter-Task Chaining further connects it to downstream, including spatial VAL and textual VAU into a cascaded chain for a unified holistic anomaly analysis.

prediction losses [Hasan et al., 2016, Thakare et al., 2022]; and *weakly-supervised* MIL frameworks used video-level tags to rank anomalous snippets [Sultani et al., 2018, Feng et al., 2021, Joo et al., 2023]. All of them need to be re-trained for unseen domains or anomaly types and provide no semantic rationale for their decisions [Ramachandra et al., 2020, Wu et al., 2024b]. To address this, *open-set* detectors emerged: OVVAD fuses LLM semantics so the system can both *detect* and *classify* novel anomalies [Wu et al., 2024a]. However, such open-set detectors still require task-specific training and provide very limited textual insight into *why* frames may be abnormal, motivating the move toward vision-language solutions with task formulations beyond temporal detection.

**VLM-based video anomaly analysis.** LAVAD [Zanella et al., 2024] introduces a fully *training-free* pipeline for temporal detection: a frozen VLM captions each frame; a prompted LLM converts the caption stream into frame-wise anomaly scores that are further refined by ensembles of foundation models. While effective when anomalies are clearly distinguishable from normality, it occasionally fails to distinguish more complex anomaly types [Ding and Wang, 2024], and lacks direct semantic explanations, providing only default VLM captions alongside computed anomaly scores. Prompt-tuning variants [Du et al., 2024, Wu et al., 2024b, Yang et al., 2024, Ye et al., 2025] optimize textual prompts to guide frozen MLLMs for certain tasks. While they reveal strong performance, they remain dependent on annotated data and deal with limited task scopes [Zhang et al., 2024b].

**Video anomaly understanding and multimodal LLMs.** With the need for deeper semantic reasoning, instruction-tuning methods such as Hawk [Tang et al., 2024] and Holmes-VAU [Zhang et al., 2024b] fine-tune VLMs on detailed, anomaly-captioned video clips to produce narrative explanations. These works have achieved more accurate descriptions but require extensive annotation and computational resources, and remain tied to seen anomaly types [Liu et al., 2025].

To sum up, we observe: strictly zero-shot methods such as Zanella et al. [2024] support temporal detection but lack spatial grounding and textual insights. Prompt-tuning variants [Du et al., 2024, Wu et al., 2024b, Ye et al., 2025] are mostly focused on only a subset of tasks/domains as the prompts are often task/domain-specific. Instruction-tuned models [Tang et al., 2024, Zhang et al., 2024b] produce rich narrative explanations, yet lack either temporal or spatial coverage and incur high annotation costs. These gaps motivate our effort to unify these tasks under a zero-shot setting.

## 3 Methodology

We show an overview of this framework in Figure 1. The video anomaly analysis task is decomposed into three major sub-tasks, as formulated in previous works, and our framework exploits the inherent

connection among them. Our unified framework can be summarized in two major components: 1) An Intra-Task Reasoning (IntraTR) extracts anomaly priors through the temporal video anomaly detection (VAD) task and then refines the temporal detection through a gated additional reasoning step. 2) Building on the reasoning process in IntraTR, an additional Inter-Task Chaining (InterTC) connects the extracted tag list and temporal score results from the initial VAD results to enable subsequent localization and understanding tasks in a cascaded manner. Detailed explanations for each component are provided in Section 3.1 and Section 3.2 respectively.

## 3.1 Intra-Task Reasoning (IntraTR) for temporal anomaly detection

**Problem formulation.** VAD can be formulated as a binary (0-1) classification at frame level. Ideally, for each input frame $f_i$, the objective is to predict an anomaly probability $s_i$. For baseline methods utilizing LLM and VLM [Zanella et al., 2024], it can be formulated as:

$$s_i = \theta_{\text{LLM}}\big(p_{\text{VAD}} \oplus \theta_{\text{VLM}}(c_i, p_{\text{caption}})\big), \qquad S_V = \big[s_1, \ldots, s_T\big], \qquad (1)$$

where $T$ is the number of frames in video $V$, $c_i$ is a short video clip representing events around frame $f_i$ and $p_{\text{VAD}}, p_{\text{caption}}$ represents prompts used respectively for video anomaly detection and clip captioning. Vector $S_V$ therefore provides a *first-pass* anomaly estimate for every frame, obtained without fine-tuning. *However, beyond this baseline, can we further leverage $S_V$ for improved reasoning?*

Trying to answer this question, our VAD pipeline treats $S_V$ not only as the final answer but also as a starting point for a structured intra-task reasoning step performed at test time. Figure 2 provided an overview of the proposed IntraTR pipeline.

**Score-guided anomaly extraction.** To identify the potential anomalies present in the video, we first conduct one forward pass producing frame-wise anomaly scores $S_V = \big[s_1, \ldots, s_T\big]$ for a video $V$ with $T$ frames, where each $s_i \in [0, 1]$. Intuitively, an anomalous event $e$ should occupy a contiguous window $W_e = \{t, \ldots, t+\ell-1\}, \ell \ll |S_V|$ reflects a local segments. Denote the mean score inside any window $W$ by $\mu(W) = \frac{1}{|W|} \sum_{j \in W} s_j$. Following the intuition that anomaly events should maintain consistently high scores, in an anomalous video, we expect to find:

$$\exists\, W_e : \ |W_e| = \ell, \ \text{such that } \mu(W_e) \geq \tau, \quad (2)$$

where $\tau$ is a natural decision boundary (e.g. $\tau = 0.5$).

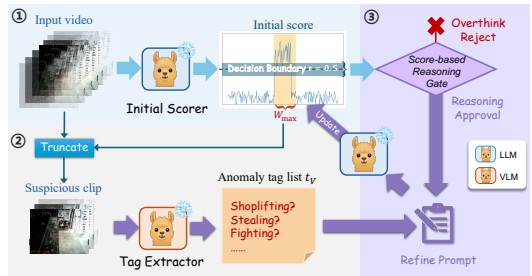

Figure 2: **Intra-Task Reasoning pipeline:** (1) the Initial Scorer produces a score curve; (2) peak detection truncates a suspicious window and the Tag Extractor generates anomaly tags $t_V$; (3) a reasoning gate refines low-confidence predictions via the Score Updater.

To find whether such a window $W_e$ exists in the video, at inference time, we slide a window of admissible length $\ell$ and select:

$$W_{\text{max}} = \arg \max_{W \subseteq \{1,\ldots,T\},\ |W|=\ell} \mu(W), \qquad (3)$$

$$\tilde{s}_V = \mu(W_{\text{max}}), \qquad (4)$$

where $W_{\text{max}}$ is the most suspicious segment and $\tilde{s}_V \in [0, 1]$ is the surrogate video-level anomaly probability. After identifying the most suspicious part of the video $V_{\text{sus}}$ indicated by $W_{\text{max}}$, we extract text contexts related to anomalies by querying VLM to generate a list of concise phrases $t_V$ summarizing the possibly related anomaly activities in the video clip $V_{\text{sus}}$ as follows:

$$V_{\text{sus}} = [f_j], \qquad j \in W_{\text{max}}, \qquad (5)$$

$$t_V = \theta_{\text{VLM}}\big(V_{\text{sus}}, p_{\text{extract}}\big). \qquad (6)$$

We then pass $W_{\text{max}}$, $\tilde{s}_V$ and $t_V$ to later stages for further processing.

**Score-based reasoning gate.** Recent studies reveal a non-monotonic trade-off between reasoning depth and accuracy in large language models: while a short chain of thought can boost performance, excessive steps often induce "over-thinking" and hallucinations [Huang and Chang, 2023, Chen et al., 2025]. Inspired by this observation, we trigger an additional reasoning pass *only* when the first-pass score is ambiguous via a score-based gate component with motivation explained below.

Starting from the raw frame scores $S_V$, we obtain the surrogate video-level probability $\tilde{s}_V$. If $\tilde{s}_V \notin [0.5-m, 0.5+m]$, the model is considered confident about its first round predictions as the prediction is positioned far from the decision boundary [El-Yaniv and Wiener, 2010]. Therefore, a gating mechanism with width $2m$ allows borderline/ambiguous videos with $\tilde{s}_V \in [0.5\pm m]$ to proceed to a second reasoning stage. With the tag list $t_V$ extracted from frames in $W_{\max}$, the task prompt is refined to $p_{\mathrm{VAD}}^* = t_V \oplus p_{\mathrm{VAD}}$.

Intuitively, $m$ quantifies the degree of *"suspicion"*, which can be either a fixed value or adaptive w.r.t. each sample. For the setting of $m$ specifically, we offer two options. It could be either 1) a fixed heuristic constant over all samples that allowing user to control the degree of suspicion, 2) or as an adaptive sample-specific variable estimated from current $V$ by $\tilde{m}_V = \mathrm{Var}(\mathrm{S_V})$ reflecting the diversion of normal/abnormal frame scores may exist in current video. We compare and discuss the impact of $m$ in Section 4.2 and Appendix B.1 correspondingly.

Based on the above, querying the scorer LLM once more with refined prompts when $\tilde{s}_V \in [0.5\pm m]$:

$$S_V^* = \theta_{\mathrm{LLM}}\big(p_{\mathrm{VAD}}^* \ \oplus \ \theta_{\mathrm{VLM}}(c_i, p_{\mathrm{caption}})\big), \qquad i = 1, \ldots, T. \tag{7}$$

The refinement yields updated frame scores $S_V^*$, replacing the initial $S_V$ for the final decision. Following established practices in prior works [Ye et al., 2025, Tran et al., 2022], we run a standard gaussian smoothing to post-process the refined $S_V$, resulting in the final $S_V^{\mathrm{pred}}$. By allocating the costly reasoning step only when the score near the margin indicates uncertainty, the method inherits the computational efficiency and robustness of selective prediction while mitigating "over-thinking" hallucinations observed in unrestricted chain-of-thought generation.

Beyond the IntraTR-assisted VAD above, we further explore leveraging the reasoning steps from the VAD task to assist downstream tasks through InterTC component in Section 3.2 and Section 3.2.

## 3.2 Inter-Task Chaining (InterTC) for holistic anomaly analysis

In this section, we cover the design of InterTC for two key sub-tasks in anomaly analysis, namely 1) spatial Video Anomaly Localization (VAL) and 2) textual Video Anomaly Understanding (VAU).

**InterTC from temporal detection to spatial localization.** Video Anomaly Localization (VAL) aims to predict spatial bounding boxes for regions in the frame $f$ containing the anomalous activities. The InterTC connects VAD with VAU using a straightforward method. Specifically, we utilized a frozen VLM $\theta_{\mathrm{LOC}}(p_{\mathrm{LOC}} \oplus f)$, guided by a base localization task prompt $p_{\mathrm{LOC}}$ for frame $f$. And then inject $t_V$ to the $p_{\mathrm{LOC}}$, producing a refined prompt $p_{\mathrm{LOC}}^*$ using a pre-defined template. Therefore, $p_{\mathrm{LOC}}^*$ is expected to be a more sample-specific and clearer guiding prompt for spatial localization and thereby improving its performance. Detailed prompt templates are included in Appendix C.1.

**Cascaded InterTC for video anomaly understanding.** Given an untrimmed surveillance video $V = (f_1, \ldots, f_T)$, video-level anomaly understanding (VAU) aims to 1) decide whether $V$ containing an abnormal event and 2) output a human-readable description $d^*$ that explains anoma-

---

**Algorithm 1:** Inter-Task Chaining prompt refinement for VAU

**Input:** video $V = [f_1, \ldots, f_T]$;
tag list $t_V$;
base prompt $p_{\mathrm{VAU}}$;
localization prompt $p_{\mathrm{LOC}}$;
surrogate anomaly score $\tilde{s}_V$;
most suspicious window $W_{\max}$
**Output:** final description $d^*$

**VAD-prior Prompt Refinement:**
  $p_{\mathrm{VAU}}^* \leftarrow t_V \oplus p_{\mathrm{VAU}}$;
**Score-gated Localization Overlay (optional):**
  **if** $\tilde{s}_V > 0.5$ **then**
    $F_{\mathrm{sel}} \leftarrow \mathrm{sample\_frames}(V, W_{\max})$;
    $bboxes \leftarrow \theta_{\mathrm{LOC}}\big(F_{\mathrm{sel}}, t_V \oplus p_{\mathrm{LOC}}\big)$;
    $V_{\mathrm{query}} \leftarrow \mathrm{draw\_boxes}(V, bboxes)$;
    **else** $V_{\mathrm{query}} \leftarrow V$;
**Final description:**
  $d^* \leftarrow \theta_{\mathrm{VLM}}\big(V_{\mathrm{query}}, p_{\mathrm{VAU}}^*\big)$;
**return** $d^*$

---

lies from the visual inputs. Formally,

$$\Theta_{\text{VAU}} : V \longrightarrow (\hat{y}_V, d^*), \qquad \hat{y}_V \in \{0, 1\}. \tag{8}$$

Unlike earlier works that train task-specific models via instruction tuning [Tang et al., 2024, Zhang et al., 2024b], our approach to $\Theta_{\text{VAU}}$ operates in a fully *zero-shot* manner. It reuses the frame-level scores $S_V$, the tag list $t_V$, and the suspicious window $W_{\text{max}}$ obtained during the earlier temporal detection and spatial localization steps to refine the anomaly understanding prompt at inference time.

Algorithm 1 provides an overview of the full *prompt refinement* step for downstream VAU task leveraging the reasoning steps from the preceding VAD and VAL tasks. Specifically, we begin with *VAD-prior Prompt Refinement* which incorporates the tag list $t_V$ from the VAD task into the anomaly description prompts, forming a more context-aware textual query $p^*_{\text{VAU}} = t_V \oplus p_{\text{VAU}}$.

Next, we apply a visual prompt enhancement called *Score-gated Localization Overlay*. Specifically, the surrogate probability $\tilde{s}_V$ *gates* a visual-prompt enhancement stage: only when $\tilde{s}_V > 0.5$. i.e. the VAD detector already believes an anomaly is present, allowing us to trust that object-level cues are meaningful and beneficial to include. For such videos we 1) sample frames inside $W_{\text{max}}$. 2) invoke a detection-capable VLM with $t_V \oplus p_{\text{LOC}}$ to obtain bounding boxes, and 3) overlay those boxes onto the corresponding frames in original video $V$, producing an annotated $V_{\text{query}}$. If $\tilde{s}_V \leq 0.5$ we skip the bounding box overlay and retain the original, unmodified video.

Finally, the VLM receives $V_{\text{query}}$ (either annotated or not) together with $p^*_{\text{VAU}}$ and outputs the description $d^*$. Since localization is performed only when the detector is confident that an anomaly exists, the inserted boxes act as reliable visual prompts rather than noisy clutters.

## 4 Experiments

### 4.1 Experimental setup

**Datasets & evaluation metrics.** We evaluate on the official test splits of three benchmarks: 1) UCF-Crime [Sultani et al., 2018] (real-world CCTV and crowd-sourced, 13 anomaly types); 2) XD-Violence [Wu et al., 2020] (800 test videos from movies, sports clips, CCTV, dashcam, cartoons); 3) UBnormal [Acsintoae et al., 2022] (211 fully synthetic surveillance videos across 29 virtual environments); 4) a more recent MSAD [Zhu et al., 2024] (14 distinct scenarios captured from various camera views, containing 360 test videos) which is less likely to overlap with pre-train data.

According to previous works, we primarily evaluate Area Under the Curve (AUC) score for the Receiver Operating Characteristic (ROC) Curve on all the datasets. Since several studies also report Average Precision (AP) on XD-Violence [Wu et al., 2020], we include AP results for reference.

Finally, for *Video Anomaly Understanding (VAU)* task, to fairly evaluate the quality of the generated $d^*$, we adopted all the video-level annotations from HIVAU-70k [Zhang et al., 2024b]. Spanning 1051 video descriptions, with 251 test videos from UCF-Crime, and 800 videos from XD-Violence, which is larger than the original video-level test set in Zhang et al. [2024b] (398 samples). In addition to traditional NLP metrics [Papineni et al., 2002, Vedantam et al., 2015, Banerjee and Lavie, 2005, Lin, 2004], we also evaluate GPT-guided scores following recent works [Tang et al., 2024, Li et al., 2024a]. More details are available in Appendix C.

**Hyperparameters & experiment details.** For VAD tasks, clip-level scoring operates on the full video with a 16-frame stride (see details in Appendix C.2). The suspicious window size for the prior extraction step is set to $\ell = \max(300, T/10)$ and fixed $m = 0.05$. We evenly subsample at most 180 frames from the window $W_{\text{max}}$ due to the limited context capacity of the VLM model to get $t_V$. As for the default VLM and LLM tested in the framework, we choose `VideoLLaMA3-7B` [Zhang et al., 2025a] and `Llama-3.1-8B-Instruct` [Grattafiori et al., 2024]. To reduce computational cost, we subsample all videos at a frame sampling stride of 16. We run all experiments on two NVIDIA GeForce RTX 3090 GPUs. Further implementation details, prompts and hyperparameter stability tests are provided in Appendix C and Appendix B.1.

Additionally, we adopted `Qwen2.5-VL-7B` [Bai et al., 2025] as the default localization VLM for VAL task, and varied different baseline VLMs including Zhang et al. [2025a], Li et al. [2024b], Bai et al.

Table 2: **VAD Performance comparison across UCF-Crime, XD-Violence, UBNormal and MSAD.** ✓ / ✗ indicate whether a method is *zero-shot* and *training-free* in terms of model parameters.

| Method | Zero-shot | Training-free | UCF-Crime AUC(%) | XD-Violence AUC(%) | XD-Violence AP(%) | UBNormal AUC(%) | MSAD AUC(%) | MSAD AP(%) |
|---|---|---|---|---|---|---|---|---|
| Sultani et al. [2018] | ✗ | ✗ | 77.92 | - | 73.20 | 50.30 | - | - |
| GODS [Wang and Cherian, 2019] | ✗ | ✗ | 70.46 | 61.56 | - | - | - | - |
| RTFM [Tian et al., 2021] | ✗ | ✗ | 83.31 | - | 77.81 | 60.94 | 86.7 | **66.3** |
| AccI-VAD [Reiss and Hoshen, 2022] | ✗ | ✗ | - | - | - | 66.51 | - | - |
| CLIP-TSA [Joo et al., 2023] | ✗ | ✗ | 87.58 | - | 82.19 | - | - | - |
| MGFN [Chen et al., 2023b] | ✗ | ✗ | 86.98 | - | 80.11 | - | 85.0 | 63.5 |
| STPrompt [Wu et al., 2024b] | ✗ | ✗ | 88.08 | - | - | 63.98 | - | - |
| OVVAD [Wu et al., 2024a] | ✗ | ✗ | 86.40 | - | 66.53 | 62.94 | - | - |
| Holmes-VAU [Zhang et al., 2024a] | ✗ | ✗ | **88.96** | - | **87.68** | - | - | - |
| MULDE [Micorek et al., 2024] | ✗ | ✗ | 78.50 | - | - | **72.80** | - | - |
| EGO [Ding et al., 2024] | ✗ | ✗ | 81.71 | - | 65.77 | - | **87.3** | 64.4 |
| AnomalyRuler [Yang et al., 2024] | ✗ | ✓ | - | - | - | 71.90 | - | - |
| VERA [Ye et al., 2025] | ✗ | ✓ | 86.55 | **88.26** | 70.54 | - | - | - |
| HolmesVAU [Zhang et al., 2024b] (ZS) | ✓ | ✗ | - | - | - | 58.54[†] | - | - |
| AnomalyRuler [Yang et al., 2024] (ZS) | ✓ | ✓ | - | - | - | 65.40[†] | - | - |
| UR-DMU [Zhou et al., 2023] (ZS) | ✓ | ✓ | - | - | - | - | 74.3 | 53.4 |
| CLIP [Radford et al., 2021] (ZS) | ✓ | ✓ | 53.16 | 38.21 | 17.83 | - | - | - |
| LLAVA-1.5 [Liu et al., 2024] (ZS) | ✓ | ✓ | 72.84 | 79.62 | 50.26 | - | - | - |
| VideoLLaMA3-7B + Llama3.1-8B (ZS) | ✓ | ✓ | - | - | - | - | 78.7 | 68.5 |
| GLM-4.1V-9B-Thinking (ZS CoT)[‡] | ✓ | ✓ | 61.80 | 72.73 | 52.93 | 60.81 | - | - |
| LAVAD [Zanella et al., 2024] | ✓ | ✓ | 80.28 | 85.36 | 62.01 | 51.06 | - | - |
| **Ours (fixed constant $m$)** | ✓ | ✓ | **84.28** | **91.34** | **68.07** | 68.98 | 85.9 | **76.4** |
| **Ours (adaptive $\tilde{m}_V$)** | ✓ | ✓ | 84.08 | 91.23 | 68.03 | **69.02** | **86.0** | 75.9 |

[†] The result is from a direct evaluation of the method trained on other non-overlapping datasets, reflecting its zero-shot performance.

[‡] Zero-shot chain-of-thought (CoT) inference VAD performance using GLM-4.1V-9B-Thinking [Team et al., 2025].

Table 3: **(a)** Ablation of inference steps. showing the effectiveness of each reasoning component. **(b)** Ablation on video-level anomaly priors. $t_{\text{oracle}}$ uses ground-truth types, and $t_V$ are actual local anomaly priors we extracted during the reasoning step.

(a) Inference component effectiveness ablations

| ① LLM-Scoring | ② Prior Reasoning | ③ Score-gated Reasoning | AUC (%) |
|---|---|---|---|
| ✗ | ✗ | ✗ | 77.67 (+0.00) |
| ✓ | ✓ | ✗ | 77.40 (-0.27) |
| ✓ | ✗ | ✗ | 80.38 (+2.71) |
| ✓ | ✓ | ✓ | **84.28** (+6.61) |

(b) Reasoning prior ablations

| Anomaly Priors | AUC (%) |
|---|---|
| $p_{\text{VAD}}$ | 81.86 |
| $\oplus\ t_{\text{oracle}}$ | 83.91 |
| $\oplus\ t_V$ **(Ours)** | **84.28** |

[2025] for VAU task. Moreover, for both VAL and VAU tasks, we leverage the anomaly priors (e.g. $t_V, \tilde{s}_V, W_{\max}$) obtained under the default configuration of IntraTR in Section 3.1.

For the *Video Anomaly Localization (VAL) task*, following previous works, we evaluate *temporal IoU (TIoU)* [Liu and Ma, 2019, Wu et al., 2024b] for each anomalous frame $f_j$ $(j = 1, \ldots, N)$, the localization head $\theta_{\text{LOC}}(f_j, p^*_{\text{LOC}})$ outputs a confidence $C_j$ and a box $B_j$. Then, the TIoU is computed as: $\frac{1}{N}\sum_{j=1}^{N}\frac{\text{Area}(B_j \cap G_j)}{\text{Area}(B_j \cup G_j)}\mathbb{I}[C_j \geq \tau]$, with $G_j$ the ground-truth bboxes, where the indicator $\mathbb{I} \in \{0, 1\}$ judges whether the confidence $C_j$ is above the default threshold $\tau = 0.5$.

## 4.2 VAD results

Table 2 summarizes the results across the three benchmarks. Across all datasets, our *zero-shot, training-free* framework outperforms the previous best zero-shot detectors by $4-6\%$ on UCF-Crime and XD-Violence, $3\%$ on UBnormal, and also generalises well on MSAD. Our method also showed competitive performance to those baselines requiring additional supervision, data or CoT reasoning steps, further proving the benefit of IntraTR component we proposed. Figure 3 qualitatively compares our method with the baseline [Zanella et al., 2024] showing significantly reduced false positive predictions. More examples are provided in Appendix D.1.

We also find that a fixed margin $m$ already performs well, although it introduces an unavoidable assumption on the test domain, while variance estimated $\tilde{m}_V$ also provides similar performances without posing any assumption on the test domain. This sample-specific "suspicion" accounts for its superiority on a synthetic dataset (UBNormal) where samples are peculiar to natural videos. We further discuss the impact of $m$ values in Appendix B.1.

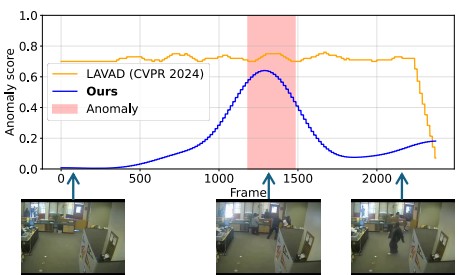

Figure 3: Anomaly scores on a video from UCF-Crime with an "Arrest" incident.

Table 4: **Comparison with previous supervised works on temporal-IoU (%) metric using zero-shot** `Qwen2.5-VL-7B`. $t_V$ comes from IntraTR, $t_{oracle}$ from ground-truth class names.

| Method | TIoU |
|---|---|
| VadCLIP [Wu et al., 2024c] | 22.05 |
| STPrompt [Wu et al., 2024b] | 23.90 |
| Qwen2.5-VL-7B (baseline) | 24.09 |
| $\oplus\ t_V$ | 25.17 |
| $\oplus\ t_{oracle}$ | **25.21** |

Table 5: **Video anomaly understanding performance comparison on two benchmark datasets.** The results are computed against ground-truth descriptions provided by [Zhang et al., 2024b]. Apart from the traditional NLP metrics (**BLEU, CIDEr, ROUGE, METEOR**), we also provide **GPT-R, GPT-D, GPT-C** metrics `Reasonability`, `Detail` and `Consistency` computed against against the ground-truth using API calls to OpenAI-GPT4.1 [OpenAI, 2025] correspondingly following previous works [Tang et al., 2024, Li et al., 2024a].

| Method | UCF-Crime [Sultani et al., 2018] | | | | | | | XD-Violence [Wu et al., 2020] | | | | | | |
|---|---|---|---|---|---|---|---|---|---|---|---|---|---|---|
| | BLEU | CIDEr | METEOR | ROUGE | GPT-R | GPT-D | GPT-C | BLEU | CIDEr | METEOR | ROUGE | GPT-R | GPT-D | GPT-C |
| InternVideo2.5-8B [Wang et al., 2025] | 0.159 | 0.011 | 0.088 | 0.103 | 0.240 | 0.266 | 0.205 | 0.209 | 0.013 | 0.119 | 0.130 | 0.456 | 0.447 | 0.433 |
| VideoChat-Flash-2B [Li et al., 2024b] | 0.165 | 0.008 | 0.108 | 0.168 | 0.488 | 0.283 | 0.404 | 0.277 | 0.026 | 0.144 | 0.186 | 0.690 | 0.576 | 0.627 |
| **+ InterTC VAU refine (Ours)** | 0.297 | 0.022 | 0.157 | 0.188 | 0.509 | 0.427 | 0.438 | 0.324 | 0.033 | 0.158 | 0.187 | 0.715 | 0.649 | 0.655 |
| VideoLLaMA3-7B [Zhang et al., 2025a] | 0.215 | 0.014 | 0.117 | 0.156 | 0.463 | 0.289 | 0.384 | 0.290 | 0.022 | 0.141 | 0.169 | 0.568 | 0.487 | 0.499 |
| **+ InterTC VAU refine (Ours)** | 0.345 | **0.023** | 0.175 | 0.188 | **0.512** | 0.428 | **0.444** | 0.399 | 0.029 | 0.198 | 0.200 | **0.721** | **0.707** | 0.668 |
| Hawk [Tang et al., 2024] † | 0.379 | 0.008 | **0.217** | 0.187 | 0.255 | **0.580** | 0.214 | 0.375 | 0.016 | 0.176 | 0.188 | 0.408 | 0.586 | 0.365 |
| HolmesVAU [Zhang et al., 2024b] † | **0.435** | 0.021 | 0.194 | **0.257** | 0.448 | 0.356 | 0.391 | 0.376 | 0.011 | 0.182 | 0.253 | 0.715 | 0.581 | **0.673** |

† Re-evaluated on our new evaluation set strictly following its default configurations.

**Ablation on test-time reasoning steps.** Table 3a evaluates the individual contributions of the three components of our inference loop. The simplest baseline, single-round direct query to a frozen VLM achieves 77.67% (`row 1`). Introducing the ① LLM-based *Scoring* component and the ② *Prior-Reasoning* step without the subsequent score-gated reasoning yields only 77.40% (`row 2`). In contrast, keeping the LLM scorer but dropping the prior reasoning module lifts performance to 80.38% (`row 3`), indicating that unrestricted "overthinking" across all samples without selective gating can conversely inject noise, causing hallucination, degrading performance. Activating all three stages, including the ③ score-gated reasoning, further raises the result to 84.28% (`row 4`), a gain of 6.61% over the raw VLM baseline. These results validate our hypothesis that confidence in anomaly presence can act as a metric to evaluate the quality of first-round prediction and therefore effectively control a proper reasoning depth for test samples.

**Ablation on $t_V$.** Table 3b isolates the effects of incorporating the textual video-level anomaly priors in the second-round reasoning for VAD. The baseline score-gated reasoning module under fixed small margin value $m = 0.05$ with an empty $t_V$ achieves a lower performance of 81.86%. Replacing the $t_V$ with ground-truth oracle class names from annotations (e.g. `''Arson, RoadAccident''`) ($t_{oracle}$) lifts performance to 83.91%, confirming that accurate anomaly priors improve detection performance. Interestingly, our automatically extracted priors $t_V$ even surpassed the oracle class names, reaching 84.28%, demonstrating that the local anomaly extraction step could effectively finalize the anomaly priors to clearer contexts than rough anomaly classes (e.g. class label "Arrest" is ambiguous, while extracted $t_V$ may include "physical altercation" which is more informative) in ground-truth. Exploiting clearer contexts leads to superior frame-wise anomaly detection performance.

## 4.3 VAL results

Table 4 shows that the zero-shot MLLM baseline already outperforms earlier supervised detectors, and that injecting anomaly tags, either from the automatically derived tag list $t_V$ in IntraTR or the ground-truth class name, yields an additional $\sim 1\%$ absolute gain in quantitative TIoU metric. Also, the tiny gap between the results using $t_V$ and the ground-truth $t_{oracle}$ suggests our $t_V$ captures near-optimal semantic cues the oracle provides, yet without requiring any manual annotation. These

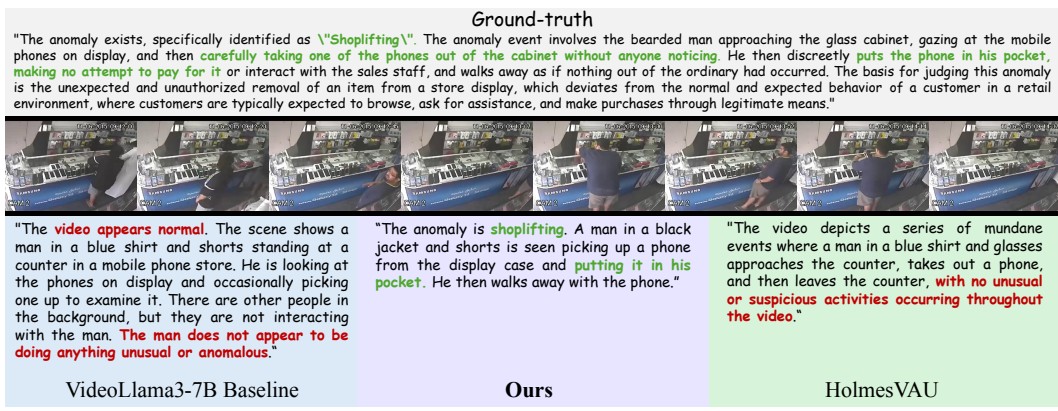

Figure 4: **Qualitative results of video anomaly understanding.** Descriptions for a video containing an incident of "Shoplifting" from different methods, where green text highlights correct descriptions/rationale about the anomaly, and red highlights statements inconsistent with the ground truth.

Table 6: Ablation study of InterTC prompt refinement steps on description quality.

| Dataset | Method | Tag $t_V$ | *bboxes* | BLEU | CIDEr | METEOR | ROUGE |
|---|---|:---:|:---:|---|---|---|---|
| UCF-Crime | ZS CoT Baseline[†] | – | – | 0.3172 | 0.0193 | 0.1651 | 0.1820 |
| | InterTC (Ablated) | ✗ | ✗ | 0.2147 | 0.0143 | 0.1167 | 0.1564 |
| | InterTC (Ablated) | ✓ | ✗ | 0.3328 | 0.0183 | 0.1684 | **0.1920** |
| | **InterTC (Full)** | ✓ | ✓ | **0.3453** | **0.0232** | **0.1750** | 0.1878 |
| XD-Violence | ZS CoT Baseline[†] | – | – | 0.3682 | **0.0381** | 0.1824 | 0.1876 |
| | InterTC (Ablated) | ✗ | ✗ | 0.2897 | 0.0219 | 0.1410 | 0.1690 |
| | InterTC (Ablated) | ✓ | ✗ | 0.3857 | 0.0270 | 0.1931 | 0.1993 |
| | **InterTC (Full)** | ✓ | ✓ | **0.3993** | 0.0288 | **0.1980** | **0.1997** |

[†] **ZS CoT**: The zero-shot VAU performance of a reasoning VLM: GLM-4.1V-9B-Thinking [Team et al., 2025], which is capable of long chain-of-thought (CoT) inference.

observations confirm that even lightweight semantic priors effectively improve spatial localization without retraining. Additional qualitative examples of localization are included in Appendix D.

## 4.4 VAU results

**Experiment results.** Table 5 compares our InterTC refinement to direct VLM inference baselines and recent instructed-tuned VAU MLLMs [Tang et al., 2024, Zhang et al., 2024b] on two different test domains, against the ground-truth description provided by HIVAU-70k [Zhang et al., 2024b]. On both domains, InterTC-refined query prompts improve the base VLM on both traditional NLP metrics and all GPT-scores (Reasonability, Detail, Consistency) of the outputs, narrowing much of the gap to instruction-tuned methods [Tang et al., 2024, Zhang et al., 2024b] and even surpassing instruct-tuned methods on several metrics. Qualitatively, we also demonstrate the descriptive capability of our framework in Figure 4. On a *shoplifting* clip, the baseline VLM [Zhang et al., 2025a] and HolmesVAU both fail to identify the abnormal act, whereas our method reports the key action ("*puts the phone in his pocket*") and labels the event as *shoplifting*. More examples are provided in Appendix D.3. These findings confirm that 1) the tag-based prompt enrichment injects crucial context and 2) localization cues further enhance narrative detail without any additional training.

**Ablation to prompt refinement steps.** To isolate the improvement of VAU metrics to each component of InterTC-assisted VAU process, we conduct corresponding ablations. As shown in Table 6, across both UCF-Crime and XD-Violence, simply enhancing the prompt with the tag list $t_V$ from VAD-priors to the base prompt accounts for the majority of the observed gains. In contrast, Inter-task chaining from the spatial localization overlay to VAU step yields a smaller, incremental lift on top of that strong improvement. We suspect primarily because the frozen VLMs have not been fine-tuned on large-scale data featuring overlaid bounding boxes, resulting in a rather marginal improvements.

While the generic "thinking" VLM [Team et al., 2025] underperforms on the more specialized VAD task (see Table 2), it performs better on VAU than zero-shot baselines. This indicates that chained inference idea adopted in both Team et al. [2025] and our InterTC can enrich textual anomaly understanding by encouraging more detailed, stepwise descriptions. However, general-purpose reasoning of Team et al. [2025] may not generalise well on the niche and complex anomaly video understanding task [Shojaee* et al., 2025], introducing content weakly related to the true anomalies. In contrast, our InterTC-guided prompts focus the description on anomaly-relevant evidence, yielding superior scores on most metrics across all video-anomaly tasks. Overall, VAD prior textual prompt refinement plays a more major role in prompt refinement, while localization visual prompts could be an optional enhancement when extra compute is available.

## 5 Conclusion

In this work, we introduced a unified, training-free framework for holistic video anomaly analysis by chaining temporal detection, spatial localization, and textual understanding in a single inference pass. Our zero-shot system consistently outperforms prior training-free baselines and approaches supervised methods across all three sub-tasks.

By structuring our pipeline as a sequence of gated reasoning steps, each sub-task enriches the next with semantic or visual priors drawn from the model's own outputs, enabling self-correction and deeper interpretability without any additional training. In video anomaly analysis specifically, where events unfold over time and space, such multi-stage inference captures structure that single-pass models miss or fail, yielding both accurate detection and user-friendly explanations without any additional training. Despite some possible limitations and potential societal impacts it may bring about a powerful yet bulky VLM system for sensitive video analysis (see more discussion in Appendix F and Appendix G), we believe this framework of treating inference as an active, context-driven process can foster more robust video analytics and may generalize to other complex vision-language tasks.

## Acknowledgements

This work was supported by the National Natural Science Foundation of China (No.92470203), Beijing Natural Science Foundation (No.L242022), the Fundamental Research Funds for the Central Universities (2024XKRC082). Jianbo Jiao is supported by an Amazon Research Award.

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

# Technical Appendices and Supplementary Material

## A    Appendix Roadmap

In this appendix, we cover the following materials:

- Additional ablation study (Appendix B)
- Additional implementation details (Appendix C)
- Additional qualitative results (Appendix D)
- Running-time Analysis (Appendix E)
- Limitations (Appendix F)
- Broader impacts (Appendix G)

## B    Additional ablation study

### B.1    Hyperparameter sensitivity tests

**Senstitivity on $m$**    We study performances under different decision-boundary-margin width values $m \in (0, 0.5)$ and dynamic $\tilde{m}_V = \text{Var}(S_V)$ presented in Table 7. Performance remains stable for $m \leq 0.2$ and $\tilde{m}_V$ and drops significantly on UCF-Crime and XD-Violence when $m = 0.4$, presumably because an overly wide margin labels many true positives as "uncertain", resulting in unnecessary hallucinations. In contrast, UBnormal [Acsintoae et al., 2022] benefits from larger $m$; the synthetic clips are originally ambiguous for pretrained models such that additional skepticism is beneficial [Yang et al., 2024]. As $m \in [0.05, 0.2]$ yields near-optimal AUC on all real-world datasets, we adopt the smallest value $m = 0.05$ as the default setting for constant $m$.

To further investigate how the IntraTR step affect model behaviours, we further visualize the density of samples with respect to the l1 distance of their video-level scores to the decision boundary $|\tilde{S}_V - \tau|$ that measures the confidence of predictions in Figure 5. Specifically, we observe that both smaller constant $m$ and the dynamic $\tilde{m}_V$ can effectively produce more high-confidence predictions, while a larger $m$ conversely results in more confusion and therefore less confident predictions overall.

**Sensitivity on window length $\ell$:**    We heuristically set our minimal suspicious window $\ell = \max(300, T/10)$, in which 300 frames is a floor for the shortest window $W_{\max}$. Since a clip $c_i$ (the smallest scoring unit) also spans 300 frames $\approx 10s$, lowering this floor has little effect.

As a result shown in Table 8, an overly large $\ell$ (as a result of a smaller divisor on video length $T$) degrades the performance. We suspect that a large window size $\ell$ hides fleeting anomalies as the

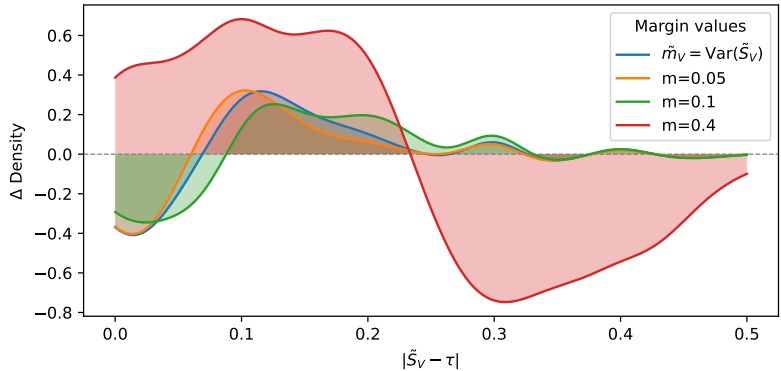

Figure 5: $\Delta$ **of Score density with regards to distance to decision boundary.** For all samples in UCF-Crime and XD-Violence, it is shown that high $m$ value resulted in more ambiguious predictions with $|\tilde{S}_V - \tau| \to 0$ while a small or local variance based $m$ effectively pushes the predictions away from decision boundary as we expected.

Table 7: **Impact of several margin values ($m \in (0, 0.5)$) on VAD performance.** All settings outperform the baseline, with stable results across different $m$ values.

| Margin values | UCF-Crime (AUC) | XD-Violence (AP) | XD-Violence (AUC) | UBNormal (AUC) |
|---|---|---|---|---|
| $m = 0.05$ | **84.28** | 68.07 | 91.34 | 68.97 |
| $m = 0.10$ | 83.10 | 68.16 | 91.40 | 69.52 |
| $m = 0.20$ | 83.57 | **68.36** | **91.60** | 70.33 |
| $m = 0.40$ | 79.21 | 67.45 | 90.81 | **70.59** |
| $\tilde{m}_V = \mathrm{Var}(S_V)$ | 84.08 | 68.03 | 91.23 | 69.02 |

Table 8: Impact of several window lengths ($\ell$) on VAD performance

| Window length | $\ell = \max(300, T/5)$ | $\ell = \max(300, T/10)$ | $\ell = \max(300, T/15)$ |
|---|---|---|---|
| UCF AUC (%) | 81.07 | **84.28** | 83.66 |

window may have higher probability of containing benign frames with lower scores, resulting in a lower estimate of the surrogate video-level anomaly probability $\tilde{s}_V$. In addition to such heuristics we used, it is also possible to introduce an additional time series segmentation model [Lovrić et al., 2014] to identify abnormal event intervals from sequences of frame scores.

**Impact of post-processing** In addition to $m$, we also evaluate the impact of the Gaussian smoothing parameter used in score post-processing. It's typical to conduct postprocessing (Gaussian, EMA) to the anomaly scores for VAD tasks [Zanella et al., 2024, Ye et al., 2025]. We followed this typical practice and implemented a simple Gaussian filter on the final score. The following Figure 6 demonstrate the robustness of our method on different $\sigma$ values we use for gaussian smoothing post-processing.

## B.2 Impact of different VLM/LLM components

**Ablation on Monolithic Multimodal LLMs** In our work, by default, we followed modular architecture of VLM + LLM from previous baseline [Zanella et al., 2024]. There are also other experiments and claims supporting this design.

In Table 3a, we have provided ablation to end-to-end VLM performance when used for scoring on every 16 frames clips. As a result, our discrete VLM, LLM framework provide better performance (84.28% against 77.67%). Which aligns with the trend reported in the baseline [Zanella et al., 2024]. We also tested a even simpler baseline of using VideoLLaMA3-7B to conduct direct end-to-end QA with complete video inputs and asking for timestamps of anomalous intervals. The Table 9 shows that such simple design gives much poorer performances even poorer overall performance.

Besides these experimental support for the modular design. Another earlier work [Chen et al., 2023a] also suggests such capability of LLMs to coordinate separate VLM models for better reasoning. Especially for cases where the task domain is a niche one under-represented in the massive pretraining data. These rationales justify our modular VLM/LLM design over single model.

**Modular Ablation on Different Multimodal LLMs** To validate the generality of our method across different MLLM components. Table 10 varies the checkpoints plugged into our pipeline. With the LLM fixed ($\theta_{\mathrm{LLM}} = \texttt{Llama-3.1-8B-Instruct}$), downgrading the vision backbone from VideoLLaMA3-7B to a 2B variant or to a Qwen2.5-VL results in only a marginal drop $\leq 1\%$ AUC, indicating that the reasoning loop compensates for weaker video features. Conversely, keeping the same VLM and swapping the LLM shows larger but still moderate drops: a 3B instruct model loses $\sim 3\%$ AUC, whereas an older Llama-2-13B loses $\sim 4\%$. Overall, every combination remains above 80% AUC, confirming the *plug-and-play* nature of our framework: it can enhance a wide range of pre-trained VLM/LLM pairs with minimal performance degradation, and it benefits most from stronger language reasoning while being relatively insensitive to vision backbone capabilities, by reducing holistic understanding into a chained process of solving simpler, modular tasks.

Furthermore, to show that our performance gain is not solely from the stronger capability of newer VLM and LLM, we run modernised baseline method [Zanella et al., 2024] under newer VideoLLama3-

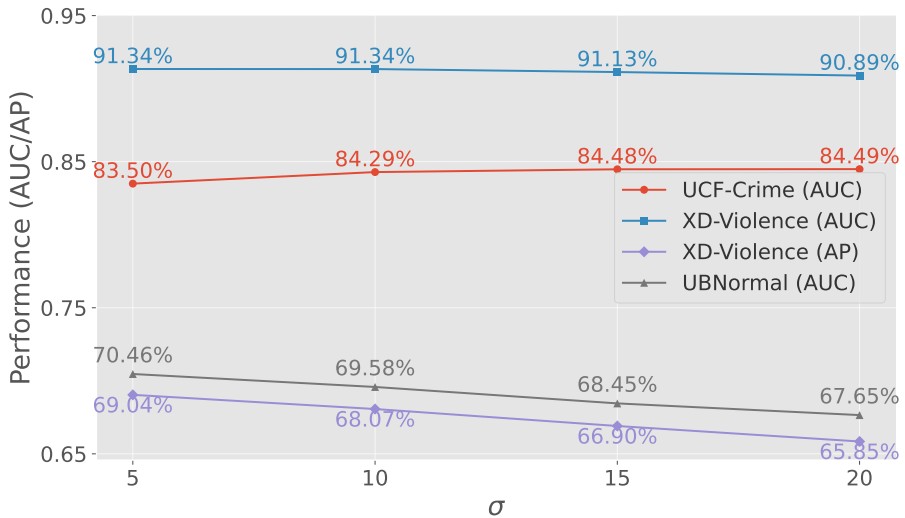

Figure 6: **VAD performance stability w.r.t. Gaussian smoothing** $\sigma$. Performance remains stable across different $\sigma$ values. We simply choose a default value $\sigma = 10$ and a SciPy's default `truncate` = `4.0` (which yields an effective radius of $4\sigma$) for all the VAD experiments.

Table 9: VideoLLaMA3-7B End-to-end VAD QA Results

| | UCF AUC (%) | XD AUC (%) | XD AP (%) | UBN AUC (%) |
|---|---|---|---|---|
| Direct QA | 58.68 | 62.52 | 33.76 | 53.73 |
| **Ours (Full)** | **84.28** | **91.23** | **68.03** | **69.02** |

7B [Zhang et al., 2025a] and Llama3.1-8B [Grattafiori et al., 2024] backbones. In Table 11, we observed a drop in single VLM performance when using newer model under Zanella et al. [2024] on UCF-Crime. This may be due to the limited capability of sentence encoding VLM [Girdhar et al., 2023], which may fail to recognise more nuanced frame caption from newer models. This problem is mitigated on XD-Violence, where more dramatic videos than mundane surveillance footage of UCF-Crime makes raw captions encoded more recognisable in the representation space.

## C   Additional implementation details

### C.1   Detailed prompts

We provide all the used prompts in this part.

**Prompts used in VAD**   Firstly, we used the same $p_{\text{caption}}$ across all datasets. Specifically:

> **Prompts for $p_{\text{caption}}$**
>
> "role": "system",
> "content": (
>       "You are an AI assistant analyzing this video segment. "
>       "Summarize the main events or actions in a concise way.
> )

As for $p_{\text{VAD}}$, we mainly adopted base prompts from Zanella et al. [2024]. Following their design, we also applied dataset priors to the prompts, as the definition of anomaly varied for each of them. Specifically, we have a base definition of anomaly events denoted as `dataset_prior` = ''`suspicious activities`''. For UCF-Crime, we change it to ''`suspicious or potentially`

Table 10: **Ablation of pretrained VLM/LLM models used on UCF-Crime.** We varied different checkpoints for the components in our framework.

| $\theta_{\mathrm{VLM}}$ | $\theta_{\mathrm{LLM}}$ | AUC (%) |
|---|---|---|
| `VideoLLaMA3-7B` | | 84.28% |
| `VideoLLaMA3-2B` | `Llama-3.1-8B-Instruct` | 83.35% |
| `Qwen2.5-VL-7B` | | 83.23% |
| | `Llama-3.1-8B-Instruct` | 84.28% |
| `VideoLLaMA3-7B` | `Llama-2-13B-Chat` | 80.70% |
| | `Llama-3.2-3B-Instruct` | 81.09% |

Table 11: Performance of "modernised" baselines [Zanella et al., 2024] with newer backbone models Zhang et al. [2025a] and Grattafiori et al. [2024].

| Method | UCF AUC (%) | XD AUC (%) | XD AP (%) |
|---|---|---|---|
| Zanella et al. [2024] (`BLIP2 FLAN-T5-XL + Llama2-13B-chat`) | 74.19 | 85.16 | 61.09 |
| Zanella et al. [2024] (`VideoLLama3-7B + Llama3.1-8B`) | 72.99 | 84.64 | 61.20 |
| **Ours** (`VideoLLama3-7B + Llama3.1-8B`) | **84.28** | **91.34** | **68.07** |

criminal'', and for XD-Violence, we opt to ``suspicious or violent'' subject to the clear definition of anomalies within each of them [Sultani et al., 2018, Wu et al., 2020]. However, on UBNormal [Acsintoae et al., 2022], where the anomalies span a wide range of spontaneous activities that may not be considered malicious by commonsense, we simply keep the base `dataset_prior`.

---

**Prompts for $p_{\mathrm{VAD}}$**

"role": "system",
"content": (
    f"How would you rate the scene described on a scale from 0 to 1, with 0 representing a standard scene and 1 denoting a scene with {`dataset_prior`} activities?"
    "Please provide the response in the form of a Python list and respond with only one number in the provided list below [0, 0.1, 0.2, 0.3, 0.4, 0.5, 0.6, 0.7, 0.8, 0.9, 1.0] without any textual explanation. It should begin with '[' and end with ']'."
)

"role": "user",
"content": (
    f"{$\theta_{\mathrm{VLM}}(c_i, p_{\mathrm{caption}})$}"
)

---

We also conducted an ablation study for the incorporation of dataset priors in Table 12, which has shown a similar trend to previous works [Ye et al., 2025, Zanella et al., 2024]. Specifically, the overall VAD performance benefited from injecting even a small context prior. Providing even brief contextual definitions of anomaly events improves baseline model performance, providing a stronger motivation for the automated extraction and utilization of the sample-specific anomaly prior we have proposed in our work.

Table 12: Ablation of dataset-level anomaly priors.

| Dataset Priors | UCF-Crime (AUC) | XD-Violence (AUC) |
|---|---|---|
| ✗ | 82.94% | 90.72% |
| ✓ | 84.28% | 91.34% |

As we described in Section 3.2, after we identified $W_{\mathrm{max}}$, we got a segment of video $V_{\mathrm{sus}}$, we queried the $\theta_{\mathrm{VLM}}$ with the $V_{\mathrm{sus}}$ and $p_{\mathrm{extract}}$ to get the tag list $t_V$.

> **Prompts for $p_{\text{extract}}$**
>
> "role": "system",
> "content": (
>     "You are an AI assistant analyzing a suspicious segment of a video. "
> )
>
> "role": "user",
> "content": (
>     f"{$V_{\text{sus}}$}"
>     "Analyze the video interval to identify any possible suspicious behaviors. "
>     "Return your answer strictly as a Python-style list of phrases that could briefly describe "
>     "the suspicious scene split by commas. "
>     "No additional commentary or text, return only the list."'
> )

To produce $p^*_{\text{VAD}}$, during inference, we augment $p_{\text{VAD}}$ prompts with a template sentence containing $t_V$. Specifically, we inject the following sentences: $\text{template}(t_V) = $ f"In addition, we have identified certain {dataset_prior} behaviors that may appear in the video. Please consider these carefully when deciding on the final anomaly rating. [Potentially reported suspicious activities: {$t_V$}]" right after the first system prompt part of $p_{\text{VAD}}$.

**Prompts used in VAL**   During spatial localization of anomaly regions in video frames, we use the simplest default prompt provided by the official release document of `Qwen2.5-VL` [Bai et al., 2025].

> **Prompts for $p_{\text{LOC}}$**
>
> "role": "user",
> "content": (
>     f"{$f_i$}"
>     "Analyze this image and identify any suspicious or anomalous region, if present."
>     "Return your answer in JSON format: "
>     "[{"bbox_2d": [x1, y1, x2, y2], "confidence": c}]."
> )

To incorporate ground-truth or extracted anomaly priors $t_V, t_{\text{oracle}}$, we simply augment the $p_{\text{LOC}}$ by adding them at the start of user prompts as hints to the model. Specifically:

> **Prompts for $p^*_{\text{LOC}}$**
>
> "role": "user",
> "content": (
>     f"{$f_i$}"
>     "The video could contain the following anomaly type: '{$t_V$}'."
>     "Localize the suspicious region or individual in this image."
>     "Return your answer in JSON format: "
>     "[{"bbox_2d": [x1, y1, x2, y2], "confidence": c}]."
> )

**Prompts used in VAU**   For VAU task, we fixed $p_{\text{VAU}}$ across different test domains (UCF-Crime, XD-violence), but varied them across different pretrained $\theta_{\text{VLM}}$ for the best baseline performance, which are:

> **Prompts for** $p_{\text{VAU}}$ (`Videochat-Flash-2B`) **[Li et al., 2024b]**
>
> user prompt = f"Please analyze the video for any anomaly activities. If there is any anomaly, describe the anomaly activities present in the video. After description, analyze why it is an anomaly without timestamps. If no anomalies are found, state that the video appears normal and then describe the scene in detail.$\{V\}$"

> **Prompts for** $p_{\text{VAU}}$ (`VideoLLaMA3-7B`) **[Zhang et al., 2025a]**
>
> "role": "system",
> "content": (
>     "You are an AI assistant analyzing a video."
> )
>
> "role": "user",
> "content": (
>     f"$\{V\}$"
>     "Please analyze the video for any anomaly activities in detail. "
>     "If there is any anomaly, describe the anomaly activities present in the video in detail. After description, analyze why it is an anomaly without timestamps."
>     "If no anomalies are found, state that the video appears normal and then describe the scene in detail."
> )

As covered in the main text, producing $p_{\text{VAU}}^*$ is simply appending template prompts with $t_V$ to the end of the system prompt ( or before the user prompt if the model does not support customizing the system prompt) of $p_{\text{VAU}}$. Specifically, $\text{template}_{\text{VAU}}(t_V) =$"For better anomaly detection and description in detail, a preliminary analysis suggests that the suspicious activity could be related to $t_V$. Use these information to guide your anomaly detection analysis.".

### C.2 Detailed sampling strategies

**Sampling clip $c_i$ around $f_i$ in VAD** Recent VLMs gain capability to process multiple frames as videos [Bai et al., 2025, Li et al., 2024b, Zhang et al., 2025a]. This is a desirable functionality we would like to exploit when dealing with frame-wise VAD. As a single frame may not be able to represent contiguous events. Therefore, following previous works sampling multiple frames to predict $s_i$ [Zanella et al., 2024, Ye et al., 2025], we opt to input a series of frames $c_i$ around the target $f_i$ instead of taking $f_i$ only. Specifically, we sample $c_i$ by following steps.

Let the video run at fps $= r_f$ and denote by $\omega$ [s] the dataset-specific temporal radius we keep on either side of $f_i$. Empirically, we set $\omega = 10$ s for UCF-Crime and XD-Violence, and $\omega = 5$ s for UBnormal (in which most clips are only 10-15 s long). The total window length in frames is $L = 2\omega\, r_f + 1$ and the half-width is $\delta = \lfloor L/2 \rfloor$. Bounding the window to the video limits,

$$a = \max\big(1,\, i - \delta\big), \qquad b = \min\big(T,\, i + \delta\big),$$

we draw $N = 10$ evenly spaced indices

$$\mathcal{I}(i) = \left\lfloor \text{linspace}\big(a,\, b,\, N\big) \right\rfloor, \qquad c_i = \{f_j \mid j \in \mathcal{I}(i)\}.$$

Thus, $c_i$ always contains 10 frames centered as much as possible on $f_i$. That means, for 30 fps videos in UCF-Crime, the sampling spans up to $\pm150$ frames (5 s) on either side, automatically shrinking near the video boundaries.

**Sampling for downstream tasks** For VAL task, we sample all the frames containing anomalies following to practice in previous works [Liu and Ma, 2019, Wu et al., 2024b]. For VAU task, we

adhere to the default configuration in Zhang et al. [2024b], which samples 16 frames per video for all the methods taking frame inputs.

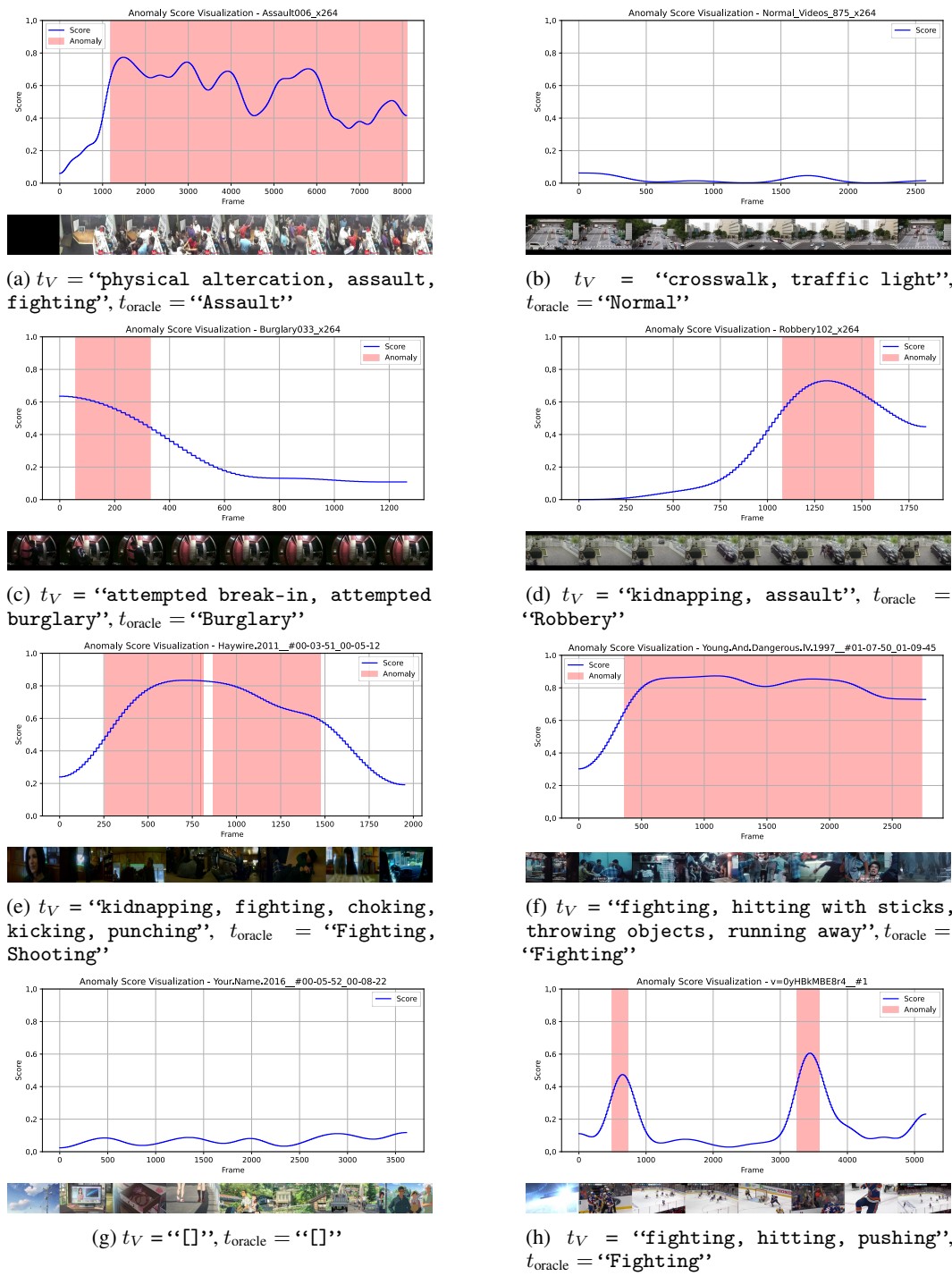

(a) $t_V$ = ''physical altercation, assault, fighting'', $t_{oracle}$ = ''Assault''

(b) $t_V$ = ''crosswalk, traffic light'', $t_{oracle}$ = ''Normal''

(c) $t_V$ = ''attempted break-in, attempted burglary'', $t_{oracle}$ = ''Burglary''

(d) $t_V$ = ''kidnapping, assault'', $t_{oracle}$ = ''Robbery''

(e) $t_V$ = ''kidnapping, fighting, choking, kicking, punching'', $t_{oracle}$ = ''Fighting, Shooting''

(f) $t_V$ = ''fighting, hitting with sticks, throwing objects, running away'', $t_{oracle}$ = ''Fighting''

(g) $t_V$ = ''[]'', $t_{oracle}$ = ''[]''

(h) $t_V$ = ''fighting, hitting, pushing'', $t_{oracle}$ = ''Fighting''

Figure 7: **Frame-wise anomaly score plots for eight representative clips.** Our method exhibit consistent performance on various video/anomaly types. The comparison between $t_V$ and $t_{oracle}$ (The original class annotated in Sultani et al. [2018], Wu et al. [2020]) is given for each sample, suggesting the qualitative performance of the anomaly prior extraction step.

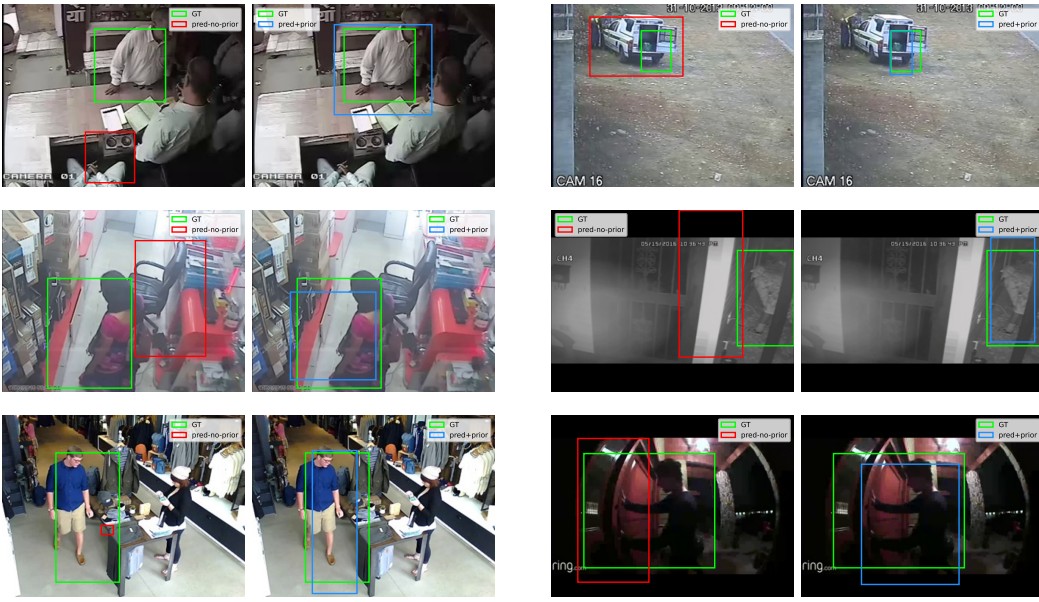

Figure 8: **Qualitative examples of our localisation outputs.** Each plot the compares detected anomaly window using baseline prompts and InterTC-refined prompts against the ground truth bounding boxes.

# D Additional qualitative results

## D.1 More results on frame-level video anomaly detection

We show additional qualitative temporal VAD results with the corresponding $t_V$ tags extracted in Figure 7. For most samples, there are clear and reasonable tags $t_V$ extracted. There are also ambiguous tags, e.g., Appendix C.2, while the performance of the VAD task remains stable. Another interesting observation here is that the $t_V$ extracted, in most cases, are analytical tags for rough $t_{oracle}$ categories. For example, in Appendix C.2, the elaborated $t_V$ = ``fighting, hitting with sticks, throwing objects, running away'' are more tractable then the rough $t_{oracle}$ = ``Fighting'', which explained the observed gap of quantitative performances when using different anomaly priors for VAD task in Table 3b.

As it is shown, despite our method exhibit decent performance in flagging various kinds of anomalies, it failed occasionally on small event gaps (e.g. in Figure 7e). We suspect that this insensitivity may be due to the uniform sampling around $f_i$ we employed to obtain $c_i$. This may result in the $c_i$ do not have the necessary granularity to represent extremely short video clips. While this is not our focus in this work, future works may consider a dynamic sampling strategy to improve the baseline VLM for VAD.

## D.2 More results on spatial video anomaly localization

The additional localization visualization in Figure 8 gives clear evidence proving the performance gain by incorporating Inter-Task Chaining of anomaly priors. The $t_V$ text prompts suggesting possible anomaly contexts allow for more accurate and reasonable groundings.

## D.3 More qualitative results on video anomaly understanding

In addition to the results shown in Figure 4, we include extra qualitative comparisons in Figure 9 and Figure 10. The results clearly show that MLLMs assisted with Inter-Task Chaining produced excellent VAU results, which accounted for the quantitative performance gains in terms of both traditional NLP metrics and preference on several dimensions of GPT-based evaluations [Tang et al., 2024]. However, we also noticed that sometimes our method produced overly verbose answers compared with other

counterparts. This actually aligns with a trend of redundant outputs discovered in LLM reasoning [Sui et al., 2025]. Despite this drawback, the majority contents in our generated descriptions are still focused on the desired topic of anomaly analysis and providing additional details, further enhancing explainability.

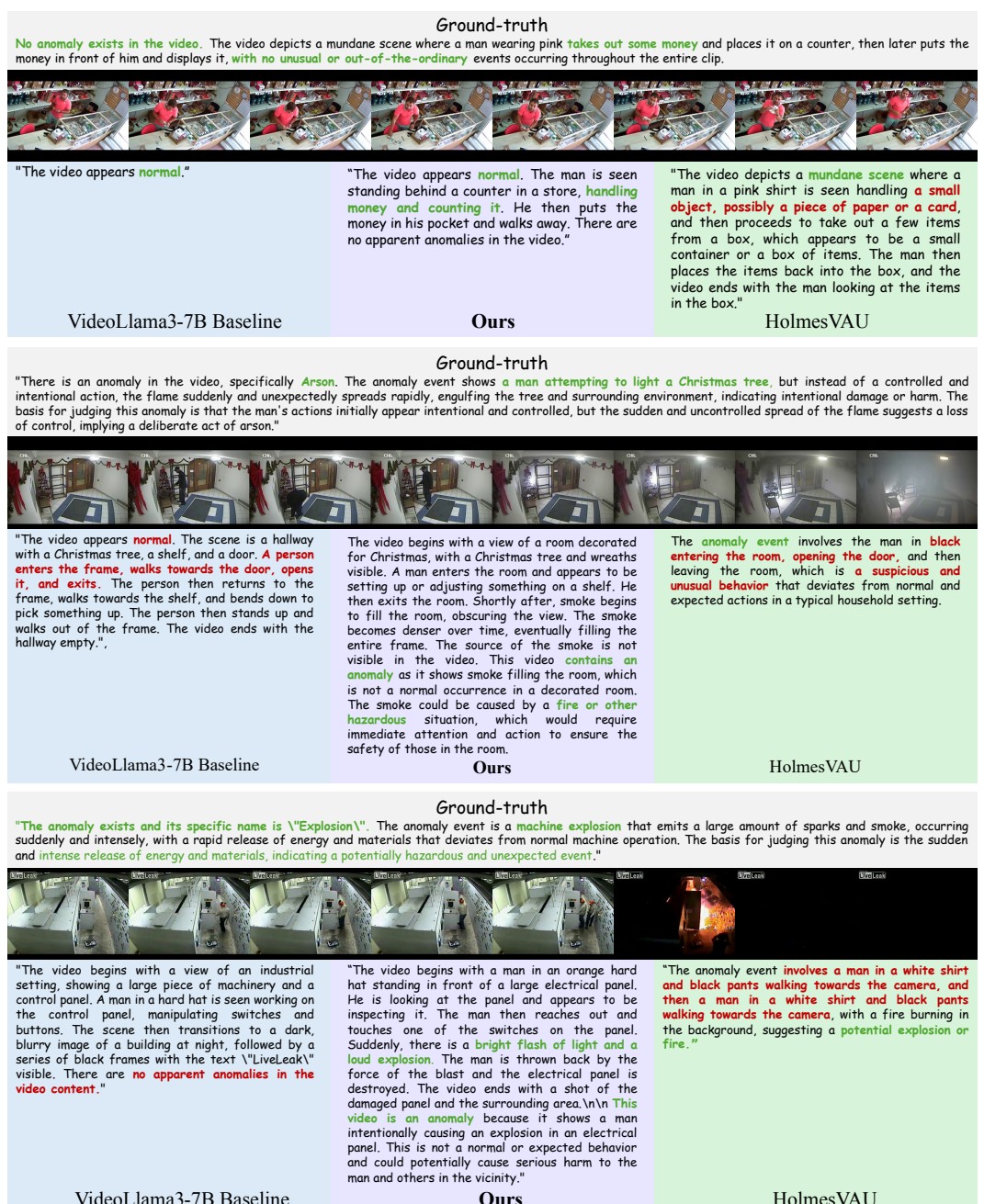

Figure 9: **Representative qualitative results for the video-anomaly understanding task (part-2).** Green parts represents correct description/reasoning about the anomaly and the red parts highlight the statements inconsistent with the ground-truth.

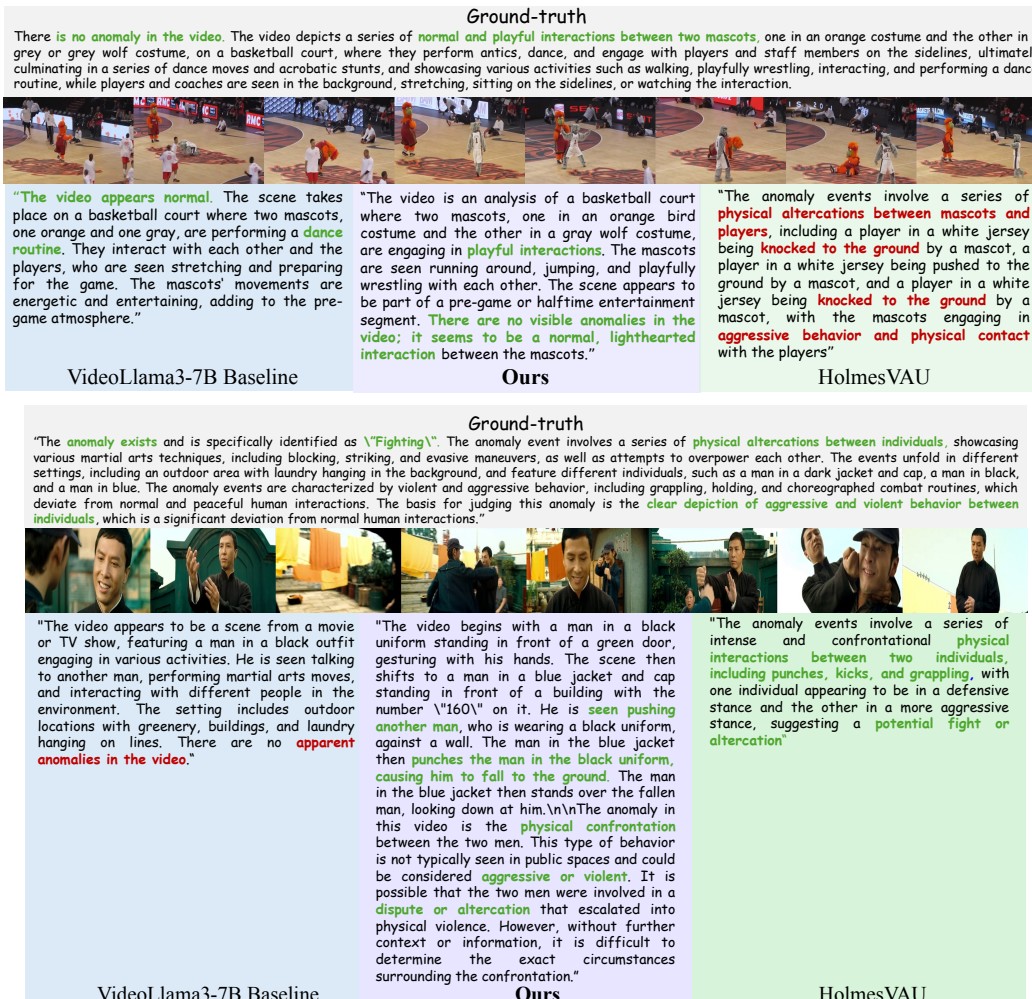

Figure 10: **Representative qualitative results for the video-anomaly understanding task (part-1).** **Green parts** represents correct description/reasoning about the anomaly and the **red parts** highlight the statements inconsistent with the ground-truth.

Table 13: **Amortised per-frame processing time (sec/frame) for a full UCF-Crime test run**, model loading excluded, smaller value means faster.

| Method | VLM Captioning | Caption Cleaning | LLM Summary | LLM Scoring | Score Refinement | VAD Overall |
|---|---|---|---|---|---|---|
| Zanella et al. [2024] | 0.06736 | 0.01490 | 0.01684 | 0.01109 | 0.00673 | 0.11691 |
| **Ours** | **0.02587** | — | — | **0.00314** | **0.00026** | **0.02927** |

# E   Running-time analysis

As we mentioned earlier in Section 3.1, our method has a relatively efficient inference process due to the selective prediction nature saving unnecessary thinking on samples where the first round scores show enough confidence. Beyond this, we also find that our method is faster than previous baseline zero-shot LLM work [Zanella et al., 2024] by design. In the following, we provide a complexity analysis of our inference steps and compare it with that of the prior work.

Our test-time IntraTR pipeline for VAD requires 1 VLM captioning query and 1 LLM scoring query per 16 frames, along with a single VLM query per suspicious video to extract tags. For videos flagged as "uncertain", we perform an additional LLM scoring query per 16 frames. In total, our method performs at most 1 VLM and 2 LLM queries per 16 frames, plus fewer than 1 VLM query per video

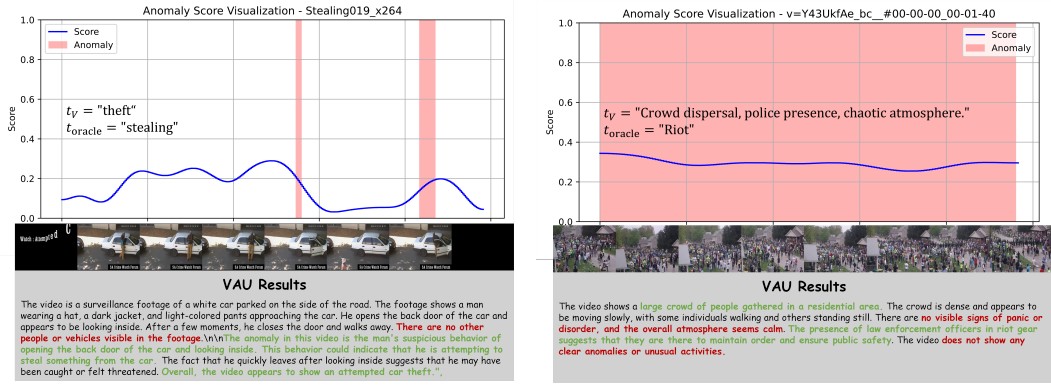

Figure 11: **Failure video anomaly analysis cases.** Both contains nuanced anomaly events that may be hard to determine. We find that for both cases, the model can still reasonable anomaly tags $t_V$ despite unsatisfactory VAD scores, therefore still yielding partially correct (**Green**/**Red** fonts represents **Correct**/**Wrong** statements) textual anomaly descriptions.

on average. In contrast, full method of previous work [Zanella et al., 2024] performs up to 5 VLM captions per frame and 2 additional LLM queries for summarising and scoring per 16 frame. It also requires additional refinement steps that introduce massive costs of encoding captions and vector searching.

Table 13 reports the amortized processing clock time inference speed on 2 RTX 3090 GPUs for a full run of the UCF-Crime test set (model loading time excluded). This gives clear supporting evidence of our efficiency advantage over the previous work.

## F   Limitations

Despite its effectiveness, our method exhibits several limitations. First, its performance is fundamentally constrained by the representational capabilities and prior knowledge of the underlying frozen multimodal large language models, which may occasionally introduce semantic biases or inaccuracies inherited from their pretraining data (see failure cases in Figure 11). Secondly, due to reliance on frozen models, our approach may suffer from reduced sensitivity in detecting highly subtle or domain-specific anomalies compared to explicitly fine-tuned models.

## G   Broader Impacts

Our work aims at enhancing public safety through better anomaly detection and interpretability in surveillance systems. However, broader deployment raises ethical considerations regarding privacy and potential misuse. Improved localization and descriptive capabilities could inadvertently facilitate invasive surveillance practices or profiling if misapplied without proper governance. Thus, any practical application of our method should be carefully regulated, ensuring transparency, accountability, and compliance with privacy laws and ethical guidelines to prevent societal harm while benefiting public security and safety.

