# OpenReview forum: "A Unified Reasoning Framework for Holistic Zero-Shot Video Anomaly Analysis"
_NeurIPS.cc/2025/Conference — NeurIPS 2025 poster_

### Official Review · Reviewer_TbPh · 2025-06-23

**Clarity:** 2
**Significance:** 2
**Originality:** 2
**Rating:** 4
**Confidence:** 4

**Summary:**

The paper addresses video anomaly detection and analysis with a training-free framework. The framework consists of two main components: Intra-Task Reasoning (IntraTR) and Inter-Task Chaining (InterTC).  In IntraTR, the authors use pre-trained VAD models and Video LLMs to generate a list of possible anomalies as refined prompts to LLMs for reasoning. In InterTC, the authors leverage VLMs to generate bounding boxes for anomalies and draw them on the frames, followed by another VLMs with refined prompts for analysis. The authors have conducted experiments on the UCF-Crime, XD-Violence and UBNormal dataset.

**Questions:**

**Important questions and concerns for the authors to address. Responses will strongly influence the final rating**
- Elaborate more on the novelty of IntraTR. Explain what the results will be if VLMs cannot capture correct suspicious behaviors (tags) or VAD baseline wrongly localizes the ambiguous segments. Providing experiments can be helpful but is not required.
- Address the concerns for Table 3 mentioned in the weakness section.
- Provide computational analysis to prove the claim in line 169.



**Optional questions and concerns for the authors to address. Good responses can increase the final rating.**
- Provide qualitative results for anomaly tags or refined prompts $p^*_{VAD}$
- Analyze what clips may not be fully captured with a fixed window size (Potential Anomaly Clips in the weakness section). Discuss possible solutions that can change the anomaly boundaries to find better anomaly segments.

Please find the relevant detail for some questions in the weakness section.

**Ethical Concerns:**

["NO or VERY MINOR ethics concerns only"]

**Final Justification:**

The authors have addressed all of my main concerns in the review, including emphasizing the novelty of IntraTR, clarifying Table 3, and providing computational analysis for the proposed method. I have been convinced that the novelty of IntraTR is not minor. Therefore, I am leaning toward increasing my final rating.

**Limitations:**

yes

**Paper Formatting Concerns:**

no major formatting issues

**Quality:**

3

**Strengths And Weaknesses:**

**Strengths:**

- The proposed method is straightforward and likely effective.
- Drawing bounding boxes on frames as clues for VLMs is interesting.
- The experimental results are promising.



**Weaknesses:**
- **Concerns about Proposed Method**
    - The novelty of IntraTR is minor because it simply takes captions from VLMs and refined prompts (anomaly tags) as inputs for LLMs to perform VAD on ambiguous clips. In addition, the design can work if the VLM is powerful enough to find anomalies in video clips and the VAD baseline is good enough to localize ambiguous segments.
- **Concerns about Performance Improvement**
    - Given that the authors adopt better VLM on captioning/tag generation (e.g., LAVAD uses BLIP-2, an image-based model, and this paper uses VideoLLaMA3-7B, a video-based model), the performance gain could come from the VLM rather than the design of the IntraTR. For example, the third row in Table 3(a) shows that using frozen VLM and LLM can achieve comparable performance (80.38%) as LAVAD (80.28%) on UCF, showing that the proposed method benefits from the strong VLM/LLM.
    - Table 3(a) does not clearly show how effective adding anomaly tags is.
- **Concerns about Experiments**
    - The component name in Table 3(a) and the paragraph (line261-271) do not match the corresponding name or appear in the method section (e.g., I could not find “soft-margin” and the authors use “Score-based reasoning gate” instead of “Score-gated Reasoning”).   In addition, the design of the simplest baseline (first row in Table 3(a)) is unclear but the performance is surprisingly good. Finally, the difference between LLM-Scoring and Prior-Reasoning is unclear because they both ask LLM to provide the scores and I did not see a reasoning prompt such as "explain what happens in the scene" in $p_{VAD}$ in line 531. The unclear table and descriptions make the reviewer hard to understand how effective each component is.
    - The authors claim that the proposed method is computation efficiency (line 169) but do not provide detailed analysis. I would like to see the computation complexity comparison between this work and other baselines.
   - In general, the refined prompt with generated tags is why VAD performance improves. Recognizing the anomaly tags for the video clips provides LLMs useful information for anomaly detection and understanding. It would be good if the authors can provide qualitative results of anomaly tags or refined prompts. Showing those results can strengthen the technical reliability of the proposed method.
- **Potential Anomaly Clips**
    - The authors set the window size to a fixed value for each video (line 237). Because the framework highly relies on the selection of anomaly clips, I wonder if some clips may not be fully captured with a fixed window size. In addition, as IntraTR only updates the prompt to refine anomaly scores, the temporal boundaries of the anomaly clips remain the same throughout testing, limiting the improvement of VAD.

---

> ### Author Rebuttal · Authors · 2025-07-30
>
> $~~~~$We thank the reviewer for providing a detailed and insightful review and recognising the strengths of this paper in methodology, ideas, and promising experimental results.
> ### Concerns about proposed method
>
> > The novelty of IntraTR is minor…the design can work if the VLM is powerful enough…the VAD baseline is good enough…
>
> $~~~~$We respectfully disagree with this claim. **IntraTR** is the **first framework that quantifies the degree of video-level uncertainty of anomaly presence** and uses it to **guide the reasoning step of the zero-shot VAD task**. Therefore, it is not **simply “caption+tags+LLM.”** The necessity of such a design is as shown by the indiscriminate, no‑gate “always‑refine” variant degrading the performance **(Table 3 (a), rows 1→2)**.
>
> $~~~~$Secondly, **IntraTR does not rely on assuming that the VLM/VAD can perfectly report anomaly segments and tags**. Sample‑specific descriptive prior tags $t_V$ extracted from ambiguous intervals can yield more accurate score estimates than "correct but coarse" ground‑truth labels, which may not reflect visible cues in inputs **(Table 3 (b), Lines 277-281)**. Thus, IntraTR’s gains stem from how priors are derived and applied, not from assuming perfect tags or exceptionally strong models.
>
> ### Concerns about Performance Improvement
>
> > ...performance gain could come from the VLM rather than the design…
>
> $~~~~$Despite simple baseline frozen caption VLM and scoring LLM with a Gaussian post-processing ($\sigma$=10) already show a comparable performance (80.38%) to the baseline. We wish to remind that **IntraTR further improves the AUC result with a large margin to 84.28%** as shown in **Table 3 (a)**.
>
> $~~~~$To show that our **performance gain is not solely from the stronger capability of newer VLM and LLM**, we run LAVAD [1] with newer VideoLLama3 and Llama3.1-8B:
>
>   |  | UCF AUC (%) | XD AUC (%) | XD AP (%) |
>   | --- | --- | --- | --- |
>   | LAVAD (BLIP2 FLAN-T5-XL + Llama2-13B-chat) | 74.19 | 85.16 | 61.09 |
>   | LAVAD (VideoLLama3-7B + Llama3.1-8B) | 72.99 | 84.64 | 61.20 |
>   | Ours (VideoLLama3-7B + Llama3.1-8B) | **84.28** | **91.34** | **68.07** |
>
> $~~~~$We observed a drop in single VLM performance when using newer model under LAVAD on UCF-Crime. This may be due to the limited capability of sentence encoding VLM (ImageBind), which may fail to recognise more nuanced frame caption differences from newer models. This problem is mitigated on XD-Violence, where more dramatic videos than mundane surveillance footage of UCF-Crime makes raw captions encoded more recognisable in the representation space.
>
> $~~~~$In addition, we show in **Table 8** that our method remains robust under much weaker 2~3B models.
>
> $~~~~$These results demonstrate the effectiveness of proposed IntraTR in providing a further boost to the baseline VLM/LLM VAD that is already strong.
>
>
> > Table 3(a) does not clearly show how effective adding anomaly tags is.
>
> $~~~~$We acknowledge that **Table 3 (a)** does not by itself reflect the effectiveness of the extracted tag list $t_V$, while **Table 3 (b) quantitatively shows** how including tag $t_V$ and ground-truth tags $t_{\text{oracle}}$ into the prompts **improves the VAD performance** by the IntraTR score-refinement.
>
>
> ### Concerns about Experiments
>
> > The component name in Table 3(a) and the paragraph (Lines 261-271) do not match the corresponding name…
>
> $~~~~$Thanks for pointing out the inconsistency, we have updated the paragraph to ensure consistent naming for components in the revised version. For now, we provide a mapping for the inconsistent names.
> - Line 263: “LLM-based Scoring” → ”LLM-Scoring“
> - Line 264: “soft-margin refinement” → “Score-gated reasoning”
> - Line 265: ”prior-rescore” → "Prior Reasoning”
>
> > …the design of the simplest baseline (first row in Table 3(a)) is unclear…
>
> The simplest baseline **inputs the VAD task prompt $P_\text{VAD}$ and 16‑frame clips directly to the VLM which outputs anomaly scores**. I.e., $S_{V}=\theta_{\mathrm{VLM}}\left(c_{i}, p_{\text {VAD}}\right), \quad i=1, \ldots, T$ in which $c_{i}$ denotes the clips and $p_{\text{VAD}}$ is standard prompt for anomaly scoring. The $S_V$ is then post‑processed by a default Gaussian smoothing with $\sigma=10$.
>
> > …the difference between LLM-Scoring and Prior-Reasoning is unclear…
>
> $~~~~$ To clarify, the **Prior Reasoning** simply runs an additional scoring step with the extracted prior tag list $t_V$ **for all videos,** which is exactly the **no‑gate “always‑refine” variant** of IntraTR.
>
> $~~~~$On the other hand, **Score-gated Reasoning** applies the Score-based reasoning gate to enable selective Prior Reasoning on “uncertain videos” as indicated by initial prediction scores (**Lines 147-172**).
>
> > I did not sefe a reasoning prompt such as "explain what happens in the scene" in $p_{\text{VAD}}$ in line 531.
>
> $~~~~$The process of making reasoning prompt $p^*_{\text{VAD}}$ that combines $p_{\text{VAD}}, t_V$ is described in **lines 545-550**. We have made this highlighted clearer in the revision.
>
> > The authors claim that the proposed method is computation efficiency (line 169) but do not provide detailed analysis.
>
> $~~~~$ Thanks for noticing this point! To clarify, the word “efficiency” in **line 169** was intended to say that the selective prediction nature of Score-based reasoning gate leads to more efficient refinement to VAD scores, as it skips samples which are far from the decision boundary and therefore reduces the average processing time.
>
> $~~~~$ However, we also noted that our method **is more efficient than the previous zero-shot baseline** LAVAD that essentially uses heavy VLM ensemble.
>
> - Our test‑time IntraTR for VAD requires one VLM captioning query and one LLM scoring query per 16 frames, plus a VLM query per suspicious video to extract tags and subsequently one LLM scoring query per 16 frames in those videos. (**< 1 VLM queries + 2 LLM queries per 16 frames overall**).
> - By contrast, [1] performs **5**  VLM captions **per frame** and **2** additional LLM queries for summarising and scoring captions **per 16 frames**, plus time-consuming refinement encoding massive numbers of sentences and retrieval.
>
> $~~~~$ To prove our advantage in terms of efficiency to previous baselines, the table below reports amortised speeds on 2 RTX 3090 for a UCF‑Crime test run:
>
>   | Methods | VLM Captioning (sec/frame) | Caption Cleaning (sec/frame) | LLM Summary (sec/frame) | LLM Scoring (sec/frame) | Score Refinement (sec/frame) | VAD Overall (sec/frame) |
>   | --- | --- | --- | --- | --- | --- | --- |
>   | LAVAD | 0.06736 | 0.01490 | 0.01684 | 0.01109 | 0.00673 | 0.11692 |
>   | **Ours** | **0.02587** | **-** | **-** | **0.00314** | **0.00026** | **0.02927** |
>
> > Provide qualitative results for anomaly tags…
>
> $~~~~$ Since links are not permitted in the rebuttal this year, we cannot include qualitative visualisations of the tags and refined prompts here. To show what the $p^*_\text{VAD}$ looks like for ordinary samples, we can provide $t_V$ for samples visualized in the current manuscript.
>
> - **Fig. 7 (a)**:  $t_V = \text{``physical altercation, assault, fighting"}$
> - **Fig. 7 (b)**:  $t_V = \text{``crosswalk, traffic light"}$
> - **Fig. 7 (c)**:  $t_V = \text{``attempted break-in, attempted burglary"}$
> - **Fig. 7 (d)**:  $t_V = \text{``kidnapping, assault"}$
> - **Fig. 7 (e)**:  $t_V = \text{``kidnapping, fighting, choking, kicking, punching"}$
> - **Fig. 7 (f)**:  $t_V = \text{``fighting, hitting, pushing"}$
> - **Fig. 7 (g)**:  $t_V = \text{``fighting, hitting with sticks, throwing objects, running away"}$
> - **Fig. 7 (h)**:  $t_V = \text{``[]"}$
>
> $~~~~$ For most samples, there are clear and reasonable tags $t_V$ extracted. There are also ambiguous tags, e.g., **Fig. 7 (b),** while the performance of VAD task remains stable. We have included further qualitative results including more failure cases in the revision.
>
> ### Potential Anomaly Clips
>
> > The authors set the window size to a fixed value for each video (line 237)…
>
> $~~~~$ We did **not** fix the window size for each video. Instead, we set a **partially adaptive $\ell = \text{max}(300, T/10)$ dependent on video length $T$**. We have revised the presentation to make it clearer.
>
> $~~~~$We would like to provide some background justification for setting a partially adaptive $\ell$. We set the floor to 300 frames ≈ 10s as the minimal “suspicious window”, because **the smallest scoring unit(clip $c_i$) also spans 10s (following previous works LAVAD, VERA), we expect that lowering this floor has little effect.**
>
> $~~~~$As for another **ceiling term $T/10$** for the window size, we tested the stability across different divisors. This provides insights to **how the window fails to capture the anomaly events**.
>
>   |  | $\ell = \text{max}(300, T/5)$ | $\ell = \text{max}(300, T/10)$ | $\ell = \text{max}(300, T/15)$ |
>   | --- | --- | --- | --- |
>   | ROC-AUC (UCF-Crime) | 81.07% | 84.28% | 83.66% |
>
> $~~~~$**An overly large $\ell$ degrades the performance**. We suspect that larger windows **hide** short-lifed anomalies (Shoplifting, Shooting, etc.) as the average of the larger windows may have higher probability of containing normal frames with lower scores.
>
> $~~~~$In contrast, **smaller windows remain stable** because the **shorter interval makes it easier to flag short-lived anomalies** with scores near the natural decision boundary.
>
> > Discuss possible solutions that can change the anomaly boundaries…
>
> $~~~~$It is possible to use an adaptive window length $\ell$. A promising extension is to introduce a **pre-trained time series model to segment the raw frame scores into intervals**, and take the one with highest average scores as $W_\text{max}$. It provides flexibility to the $\ell$ without posing assumptions on the target domain. We have added this to **Section E** in the revision.
>
>
> $~~~~$We thank the reviewer again for the insightful review, which helped us improve the paper. And we are happy to address any further concerns.

---

> > ### Comment · Reviewer_TbPh · 2025-08-04
> >
> > I appreciate the detailed response from the authors. I am convinced that the design of IntraTR is not minor. Specifically, the “overthinking” or “always-refining” design using LLMs can harm the performance of VAD, which is a noteworthy and interesting finding. In addition, the authors have shown that the proposed method achieves better efficiency than the baseline. Therefore, I am leaning toward increasing my final rating.

---

> ### Author Response · Authors · 2025-08-05
>
> Thank you for your thoughtful review and constructive feedback. Your comments helped us clarify the novelty of IntraTR, and provide additional analysis on efficiency and qualitative results. These changes have been updated to the manuscript。 We're glad the revisions addressed your concerns and appreciate your recognition of the contribution of our work. We are happy to address any further concerns.

---

### Official Review · Reviewer_DJMu · 2025-06-30

**Clarity:** 2
**Significance:** 3
**Originality:** 3
**Rating:** 4
**Confidence:** 4

**Summary:**

This paper proposes a unified, training-free reasoning framework for video anomaly analysis, aiming to integrate three key tasks: Video Anomaly Detection (VAD), spatial localization (VAL), and semantic understanding (VAU), thereby enabling comprehensive zero-shot anomaly analysis. Unlike previous methods that focus solely on temporal detection or a single sub-task, this framework employs a chained task reasoning mechanism that sequentially infers "when, where, and why" anomalies occur during inference—without any additional training or fine-tuning.

**Questions:**

See in the weaknesses section.

**Ethical Concerns:**

["NO or VERY MINOR ethics concerns only"]

**Final Justification:**

The rebuttal clarified key concerns, including spatial annotations, segment-level scoring, and model usage. While the Gaussian smoothing detail remains insufficient, the overall clarity and motivation of the paper convinced me to raise my score from Borderline Reject to Borderline Accept.

**Limitations:**

yes

**Quality:**

3

**Strengths And Weaknesses:**

Strengths：
1. This paper is well-motivated, as it tackles the anomaly reasoning task under a training-free paradigm.
2. This paper is easy to read.

Weaknesses：
1. The authors claim to address the sub-task of spatial understanding; however, I could not find any quantitative evaluation of spatial performance in the main paper. Moreover, in the qualitative visualizations provided in the appendix, it is unclear where the ground-truth annotations for the spatial data come from.
2. The authors apply Gaussian smoothing as a post-processing step, but I have two concerns. In addition to the standard deviation $\sigma$, Gaussian smoothing also involves a window size parameter, which is not specified in the paper. Furthermore, the authors do not conduct a sensitivity analysis regarding the choice of window size.
3. How do the authors obtain anomaly scores for every segment of the video if they only sample 180 frames?
4. According to the methodology, the authors use both VLMs and LLMs, but the framework diagram does not clearly indicate which specific model is used at each stage.

---

> ### Author Rebuttal · Authors · 2025-07-30
>
> We would like to thank the reviewer for the careful read and for recognising the motivation and readability of our work. Below, we address each of the weaknesses you raised.
>
> -----
> ### Lacking evaluation of spatial performance
>
> >  ...I could not find any quantitative evaluation of spatial performance in the main paper...it is unclear where the ground‑truth annotations for the spatial data come from.
> >
> $~~~~$**Table  4** reports *Temporal Intersection‑over‑Union* (**TIoU**), a metric introduced in [1] **specifically for spatial grounding**. The bounding boxes come from the officially released UCF‑Crime annotations in [1] as well. **Lines 224‑228** explain both the metric and the **source of bounding‑box annotations from [1] we use**. To improve visibility, we have referenced the definition at first mention in **Section 4** in the revised version.
>
>
>
> ### Missing window size parameter for Gaussian smoothing
>
> > “Gaussian smoothing also involves a window‑size parameter, which is not specified, and the authors do not conduct a sensitivity analysis regarding the choice of window size.”
> >
>
> $~~~~$Thank you for pointing this out. To clarify, we use **the default SciPy implementation** `scipy.ndimage.gaussian_filter1d()`providing the $\sigma$ values and leave all other parameters at their defaults following previous works [2-4]. The effective radius (window size) therefore equals $\mathrm{round}(\texttt{truncate}\times\sigma)$ with the default `truncate = 4.0`. This default setting was used in all experiments. We have updated the manuscript to make this explicit.
>
> ### Concerns on method clarity
>
> > “How do the authors obtain anomaly scores for every segment of the video if they only sample 180 frames?”
> >
> $~~~~$We wish to clarify that, as stated in **Lines 236-238**, we subsample at most 180 frames in the $W_{max}$ to get the anomaly prior $t_V$. We **do not impose this upperbound of 180 frames during frame scoring step** to get the frame scores $s_i$.
>
> $~~~~$Instead, as described in **Lines 116-121**, for every 16 frames, we sample a clip centred on the index to compute the caption (see sampling details in **Section C.2, Lines 569-582**), and score each clip by querying the LLM with the captions. The output scores therefore represent initial anomaly scores per 16-frame periods. After this initial scoring, we run IntraTR refinement steps as specified in **Lines 164-172** to refine the frame anomaly scores. We have adjusted the presentation and added the clarifications in the revised paper.
>
> -----
>
> > “The framework diagram does not clearly indicate which specific model is used at each stage.”
> >
> - For the VAD task, the VLM is used for initial captioning and then to produce the video‑level anomaly tag $t_V$ (**Line 145**). The LLM then works for the initial scoring on clip captions and an additional Intra-Task Reasoning (IntraTR) to selectively refine the scores with $t_V$ augmented prompt $p^*_\text{VAD}$ (**Lines 164-172**).
> - In our main experiments, we used VideoLLama3-7B as the default VLM and Llama3.1-8B as the default LLM (see **Lines 239-240**). We also tested other models in **Table 8.**
> - In the proposed Inter-Task Chaining (InterTC) framework, with the video‑level anomaly tag $t_V$ extracted in IntraTR steps, the subsequent localisation/understanding task can input the sample with tag $t_V$ augmented prompts $p^*_{\text{task=VAL,VAU}}$  to any VLMs (see details covered **Section 3.2**). We varied the VLMs used for downstreams in **Table 5**.
>
> $~~~~$To improve clarity, we have modified the blocks in **Fig. 2** and added more detailed in-text descriptions in the revised version.
>
> -----
>
> $~~~~$We hope these responses address your concerns, thank you again for the insightful review, and we are happy to clarify any further points.
>
> -----
>
> [1] Kun Liu and Huadong Ma. Exploring background-bias for anomaly detection in surveillance videos.383 In Proceedings of the 27th ACM International Conference on Multimedia, pages 1490–1499, 2019.384
>
> [2] Acsintoae, Andra, et al. "Ubnormal: New benchmark for supervised open-set video anomaly detection." Proceedings of the IEEE/CVF conference on computer vision and pattern recognition. 2022.
>
> [3] Flaborea, Alessandro, et al. "Multimodal motion conditioned diffusion model for skeleton-based video anomaly detection." Proceedings of the IEEE/CVF international conference on computer vision. 2023.
>
> [4] Micorek, Jakub, et al. "Mulde: Multiscale log-density estimation via denoising score matching for video anomaly detection." Proceedings of the IEEE/CVF Conference on Computer Vision and Pattern Recognition. 2024.

---

> > ### Comment · Reviewer_DJMu · 2025-08-04
> >
> > I sincerely appreciate your detailed rebuttal. It clarified my concerns, and I will raise my score accordingly.

---

> ### Author Response · Authors · 2025-08-05
>
> Thank you for your time and detailed review on our submission. The feedbacks have helped us improve the clarity on the metrics, hyperparameters and methodologies, and we have updated the manuscript accordingly. We appreciate your thoughtful comments and are happy to address any further concerns.

---

### Official Review · Reviewer_jCKt · 2025-07-02

**Clarity:** 3
**Significance:** 2
**Originality:** 2
**Rating:** 4
**Confidence:** 4

**Summary:**

The paper presents a Unified Reasoning Framework for Holistic Zero‑Shot Video Anomaly Analysis, which leverages frozen vision–language models (VLMs) and large language models (LLMs) to perform, without any additional training, three core anomaly tasks: Temporal Video Anomaly Detection (VAD) via an Intra‑Task Reasoning (IntraTR) pipeline that computes frame‑wise anomaly scores, extracts contextual “anomaly tags”.Spatial Video Anomaly Localization (VAL) by chaining the VAD priors (tag list and refined scores) into a frozen VLM localization head, yielding bounding boxes for anomalous regions;Textual Video Anomaly Understanding (VAU)

**Questions:**

**Suggestions**- Overall, the quality of the work can be improved by comparing the proposed zero-shot technique with other prompting methods, especially Chain-of-Thought (CoT) and recently released reasoning-based models. In addition, the issue of data contamination also needs to be addressed to fully validate the zero-shot performance of the proposed method.
- **Zero‑Shot CoT Baselines**  Addition of chain‑of‑thought (CoT) baseline would help.

- **Data Contamination** :- To avoid data contamination issues authors could provide results using LLM/VLM  released before a dataset.

- **Ablate Monolithic Multimodal LLMs**   Evaluate an end‑to‑end GPT‑4V or VideoLLaMA pipeline without discrete VLM/LLM modules.

**Ethical Concerns:**

["NO or VERY MINOR ethics concerns only"]

**Final Justification:**

The authors have addressed key concerns raised during the review. They added new experiments, clarified architectural choices, and provided evidence supporting the generalization capabilities of their approach. While some aspects could be further explored, the paper is significantly strengthened, and I have increased my score accordingly.

**Resolved Issues:**
- Ran preliminary evaluation on the recent MSAD benchmark to support zero shot generalization claims and removing any chance of data contamination.
- Justified the reasoning behind using a modular LLM-VLM pipeline instead of a monolithic VLM.

**Unresolved Issues**
- CoT baseline still missing.

**Limitations:**

yes

**Quality:**

2

**Strengths And Weaknesses:**

**Strengths**
- This work presents a novel zero-shot approach for anomaly detection
- Method validation across multiple anomaly detection formsFramework demonstrates functionality on various modes of anomaly detection like VAU,VAD and VAL
-Testing conducted on three benchmarks (CCTV, dash-cam/movie violence, synthetic scenes) with performance metrics compared against existing methods. Results show 91.3% AUC on XD-Violence versus 85.4% for baseline approaches.
-Systematic ablation analysis - This work examines individual component contributions including LLM scoring, tag priors, gated refinement, and fixed versus adaptive gating margins.

**Weaknesses**
- Absence of zero-shot CoT baseline comparisons No empirical evaluation against alternative zero-shot chain-of-thought approaches.
- Dataset contamination UCF-Crime, XD-Violence, and UBnormal datasets may overlap with model pretraining data. Evaluation lacks recent benchmarks or challenging scenarios that would validate generalization claims. Authors could refer to more recent datasets like MSAD[1]
- Insufficient justification for modular architecture : Separation of vision (VLM) and reasoning (LLM) components lacks empirical support. No comparison with end-to-end multimodal models (GPT-4V, VideoLLaMA) to assess performance trade-offs or computational efficiency.
[1] Advancing Video Anomaly Detection: A Concise Review and a New Dataset.Liyun Zhu, Lei Wang, Arjun Raj, Tom Gedeon, Chen Chen

---

> ### Author Rebuttal · Authors · 2025-07-30
>
> $~~~~$We thank the reviewer for the detailed and helpful review, which acknowledged the novelty, versatility and completeness of our paper. Below, we address the concerns raised:
>
>
> ### Zero‑Shot CoT Baselines
>
> > Addition of chain‑of‑thought (CoT) baseline would help.
>
> $~~~~$We did not find existing Chain-of-Thoughts (CoT) prompts that are customised for reasoning steps in video anomaly-related tasks. Therefore, to test whether reasoning steps can help, we chose to test the performance of a recent vision reasoning model GLM-4.1V-9B-Thinking (released Jul. 2025) that is claimed to be **a VLM capable of long chain-of-thought (CoT) inference**.
>
> - Results:
>    - Temporal Video Anomaly Detection (VAD) performance:
>     |  | UCF AUC(%) | XD AP (%) | XD AUC (%) | UBN AUC (%) |
>     | --- | --- | --- | --- | --- |
>     | GLM-4.1V-9B-Thinking (per-16-frame scoring) | 61.80 | 72.73 | 52.93 | 60.81 |
>     | **Ours** | **84.28** | **91.34**  | **68.07**  | **68.98** |
>
>    - Spatial Video Anomaly Localisation (VAL) performance:
>
>      |  | TIoU(%) |
>      | --- | --- |
>      | GLM-4.1V-9B-Thinking | 10.33 |
>      | Qwen2.5-VL-7B (baseline) | 24.09 |
>      | **Qwen2.5-VL-7B (with $t_V$)** | **25.21** |
>
>   - Textual Video Anomaly Understanding (VAU) performance:
>     - On UCF-Crime:
>
>
>         |  | BLEU | CIDEr | METEOR | ROUGE |
>         | --- | --- | --- | --- | --- |
>         | GLM-4.1V-9B-Thinking | 0.317 | 0.019 | 0.165 | 0.182 |
>         | **Ours** | **0.345** | **0.023**  | **0.175**  | **0.188** |
>     - On XD-Violence:
>
>
>         |  | BLEU | CIDEr | METEOR | ROUGE |
>         | --- | --- | --- | --- | --- |
>         | GLM-4.1V-9B-Thinking | 0.368 | **0.038** | 0.182 | 0.187 |
>         | **Ours** | **0.399** | 0.029 | **0.198** | **0.200** |
>
> $~~~~$From these observations, despite the reasoning model achieves a **comparable performance on the VAU task, we noticed that it is not working well on more specialised tasks** of temporal VAD and spatial VAL. Especially for the localisation task, we find that the thinking model reached a much lower TIoU than earlier non-reasoning Qwen2.5-VL-7B baseline. This is expected since the reasoning steps the model has learned may **not generalise to the specific tasks** requiring fine-grained understanding of video anomalies.
>
> $~~~~$On the other hand, the **reasoning steps for VAU can help the model produce more details about the video scene** (Fig. 1 (a) in [1] reported that reasoning models tend to generate much longer responses.) The excessive details described may explain the performance improvement over non-thinking models on the VAU task.
>
>
> ### Data Contamination
>
> > Dataset contamination UCF-Crime, XD-Violence, and UBnormal datasets may overlap with model pretraining data.
> >
>
> $~~~~$We respectfully **disagree with this claim as there is no clear evidence for such overlap**. For all the VLM/LLM we use, none reports the use of niche VAD dataset/tasks. (See Table 1-3 in [2], Section 2.2.1 of [3] and Section 8 of [4].) Therefore, we trust those models were not trained on the VAD test samples.
>
> $~~~~$In addition, even if a handful of test examples were seen during pre‑training, given that none of them were tuned to complete VAD task, they **do not have access to the anomaly groundtruth** (temporal intervals, bounding boxes and textual anomaly descriptions). Therefore, they **lack prior knowledge of the video anomalies** beyond the “commonsense” inherited from language model components in it.
>
> -----
>
> > Evaluation lacks recent benchmarks or challenging scenarios that would validate generalization claims...MSAD
>
> $~~~~$Thanks for the suggestion. We also considered this MSAD dataset and requested access before our initial submission, but we did not receive a response at that time. Following the suggestion, we requested the dataset again during the rebuttal period and have now been granted access.
>
> $~~~~$We managed to run some **prelinminary evaluation results as shown in table below** (the first 4 rows are results taken from [5, 6]), which shows our method **achieved comparable performance to SoTA** weakly supervised (WS) baselines. We are currently running additional experiments and will include further zero-shot baseline comparisons in the revised manuscript.
>
> | Methods | MSAD AUC (%) | MSAD AP (%) |
> | --- | --- | --- |
> | *WS RTFM* | 86.7 | 66.3 |
> | *WS EGO* [5] | **87.3** | 64.4 |
> | ZS MGFN | 61.8 | 31.2 |
> | ZS UR-DMU | 74.3 | 53.4 |
> | ZS VideoLLama3-7B (end-to-end) | 63.9 | 37.6 |
> | ZS VideoLLama3-7B + Llama3.1-8B | 78.7 | 68.5 |
> | **Ours (fixed constant $m$)** | 85.9 | **76.4** |
> | **Ours (adaptive $m$)** | 86.0 | 75.9 |
>
> $~~~~$Also, given that UCF-Crime (2018), XD-Violence (2020) and UBNormal (2021) were released earlier than the emergence of large VLMs, we can hardly find any strong VLM models that are released before them. However, we wish to clarify that the ground-truth description we tested the VAU task on comes from HIVAU-70k [7], which is a **very recent benchmark released in Jan. 2025**. Making it **unlikely** to be included in the pre-training set of the models picked (7b, VideoChat-Flash, Qwen 2.5 VL all had their first release in Jan. 2025).
>
> ### Ablate Monolithic Multimodal LLMs
>
> > Insufficient justification for modular architecture: Separation of vision (VLM) and reasoning (LLM) components lacks empirical support. No comparison with end-to-end multimodal models (GPT-4V, VideoLLaMA)...
>
> $~~~~$We **followed modular architecture of VLM + LLM from previous baseline**, LAVAD [8]. There are also other experiments and claims supporting this design:
>
> $~~~~$ In the manuscript, we have provided **ablation to end-to-end VLM performance** when used for scoring on every 16 frames clips in **Table 3 (a).** As a result, our discrete VLM, LLM framework provide better performance (84.28% against 77.67%).
>
> $~~~~$Same trend was also reported in Table 3 of [8] in which the performance dropped from 80.28% to 72.70% when removing the LLM-scoring module.
>
> - Another earlier work [9] also suggests such **capability of LLMs to coordinate VLM models for better reasoning**. Which could be reasons for degragations observed above when removing the LLMs.
> - Following the suggestion, we also attempted to use VideoLLaMA3-7B direct end-to-end QA with complete video inputs and asking for timestamps of anomalous intervals, which gives even poorer overall performance:
>
>   |  | UCF AUC (%) | XD AUC (%) | XD AP (%) | UBN AUC (%) |
>   | --- | --- | --- | --- | --- |
>   | Direct QA | 58.68 | 62.52 | 33.76 | 53.73 |
>   | **Ours** | **84.28** | **91.23** | **68.03** | **69.02** |
>
> $~~~~$These rationales justify our modular VLM/LLM design over single model. We have included this discussion in the revised paper. Also, the sub-optimal end-to-end performance suggests that the VLM/LLM may still underfit on video-anomaly related tasks originally, which makes the Data Contamination appears to be less likely.
>
> -----
>
> $~~~~$Thank you for your insightful review and suggestions, which have helped us improve the paper. We hope our answer could address your concern. And we are happy to clarify any further concerns.
>
> -----
>
> [1] Chen, Xingyu, et al. "Do not think that much for 2+ 3=? on the overthinking of o1-like llms." arXiv preprint arXiv:2412.21187 (2024).
>
> [2] Zhang, Boqiang, et al. "Videollama 3: Frontier multimodal foundation models for image and video understanding." *arXiv preprint arXiv:2501.13106* (2025).
>
> [3] Bai, Shuai, et al. "Qwen2. 5-vl technical report." *arXiv preprint arXiv:2502.13923* (2025).
>
> [4] Li, Xinhao, et al. "Videochat-flash: Hierarchical compression for long-context video modeling." *arXiv preprint arXiv:2501.00574* (2024).
>
> [5] Ding, Dexuan, et al. "Learnable Expansion of Graph Operators for Multi-Modal Feature Fusion." *The Thirteenth International Conference on Learning Representations*.
>
> [6] Zhu, Liyun, et al. "Advancing video anomaly detection: A concise review and a new dataset." *Advances in Neural Information Processing Systems* 37 (2024): 89943-89977.
>
> [7] Zhang, Huaxin, et al. "Holmes-vau: Towards long-term video anomaly understanding at any granularity." *Proceedings of the Computer Vision and Pattern Recognition Conference*. 2025.
>
> [8] Zanella, Luca, et al. "Harnessing large language models for training-free video anomaly detection." *Proceedings of the IEEE/CVF Conference on Computer Vision and Pattern Recognition*. 2024.
>
> [9] Chen, Liangyu, et al. "Large language models are visual reasoning coordinators." *Advances in Neural Information Processing Systems* 36 (2023): 70115-70140.

---

> ### Comment · Reviewer_jCKt · 2025-08-04
>
> I would like to thank the authors for the detailed and thoughtful response, as well as for conducting the additional experiments.
> ### On Chain-of-Thought (CoT) Baselines
> The inclusion of a reasoning-capable model like GLM-4.1V-9B-Thinking is a good step . That said, have the authors considered adapting the CoT prompt style specifically for VAD/VAL/VAU tasks when using base VLMs
>
> ### On Data Contamination
> I understand the authors’ rationale and appreciate the response. However, my point was less about known overlap and more about uncertainty since many VLMs do not disclose full pretraining mixture, it is difficult to rule out overlap with older datasets like UCF-Crime or XD-Violence. Evaluating on newer benchmarks such as MSAD, as the authors have now done, is a deterministic way to strengthen the zero-shot claims.
>
> Overall, incorporating these experiments and clarifications significantly improves the paper, both in completeness and in rigor. I am inclined to raise my score based on this improved version.

---

> ### Author Response · Authors · 2025-08-05
>
> Thank you for your detailed and constructive review. Following your suggestions, we included a reasoning-capable CoT model baseline, addressed dataset contamination concerns (by discussion and conducted evaluation on a newer dataset), and added comparisons with end-to-end multimodal models. We agree that these additions have improved the clarity and robustness of our work. We appreciate your input in strengthening the paper.

---

> ### Author Response · Authors · 2025-08-05
>
> Below, we wish to clarify your remaining concerns on the CoT baselines:
>
> > have the authors considered adapting the CoT prompt style specifically for VAD/VAL/VAU tasks when using base VLMs?
>
> As we explained in the rebuttal, **we did not find any existing CoT prompt templates specifically designed** for VAD/VAL/VAU tasks. **Designing or optimizing such prompts falls slightly outside the scope of our paper**. Moreover, prior work has reported **minimal performance variation when applying different prompt templates** without task-specific method design (see Table 4 in [1], where performance remains relatively stable). Therefore, a baseline CoT prompt simply adding ```"Think step by step."``` is not considered for our experiments.
>
> Additionally, we would like to mention that, among the other baselines we evaluated, **one prompt-based reasoning approach, AnomalyRuler [2]**, offers insight into how CoT-style reasoning might benefit base VLMs. AnomalyRuler [2] learns rules from a training set to enhance the prompts with the knowledge normal/abnormal behaviours (**a step somewhat analogous to few-shot CoT prompting**). It have shown **lower** zero-shot (when rules learned from a different domain) performance to our method as indicated by **Row 15 of Table 2**.
>
>
> Thank you for your insightful review and suggestion again, we are happy to address any further concerns.
>
> -----
>
> [1] Zanella, Luca, et al. "Harnessing large language models for training-free video anomaly detection." Proceedings of the IEEE/CVF Conference on Computer Vision and Pattern Recognition. 2024.
>
> [2] Yang, Yuchen, et al. "Follow the rules: Reasoning for video anomaly detection with large language models." European Conference on Computer Vision. Cham: Springer Nature Switzerland, 2024.

---

### Official Review · Reviewer_Savy · 2025-07-05

**Clarity:** 3
**Significance:** 3
**Originality:** 3
**Rating:** 4
**Confidence:** 4

**Summary:**

This paper presents a unified reasoning framework for video anomaly analysis. By leveraging intra-task reasoning and inter-task chaining, the approach integrates temporal detection, spatial localization, and semantic explanation into a unified framework, improving  both interpretability and generalization. The framework achieves state-of-the-art zero-shot performance on multiple benchmarks, demonstrating the potential of foundation models, when guided by well-designed prompts, to deliver practical and interpretable video anomaly analysis.

**Questions:**

1. How sensitive is the method to the choice of hyperparameter l, given the varying duration of anomaly events? A brief analysis would be helpful.

2. In Table 2, why does adaptive m underperform compared to fixed m? Please explain the possible reasons.

3. What is the inference speed of the full test-time pipeline involving multiple VLMs? Any runtime or resource analysis would be useful.

**Ethical Concerns:**

["NO or VERY MINOR ethics concerns only"]

**Final Justification:**

Thank you for the thoughtful rebuttal. I will be maintaining my current score.

**Limitations:**

The paper does not explicitly discuss the limitations or potential societal impacts of the proposed method. The authors are encouraged to:

Discuss the computational cost and inference latency, especially given the use of multiple VLMs at test time.

Address possible failure cases, such as ambiguity in anomaly definition or limitations in zero-shot generalization to unseen domains.

**Quality:**

3

**Strengths And Weaknesses:**

Strengths：
1. The paper proposes a unified framework that seamlessly integrates multiple tasks in video anomaly analysis, including detection, localization, and explanation.
2. The approach achieves state-of-the-art performance in a zero-shot setting, without requiring any additional training or fine-tuning, which highlights its strong generalization capability.

Weaknesses：
1. Since anomaly events can be either very brief or span extended durations, how sensitive is the method’s performance to the choice of hyperparameter l? A discussion or analysis on this would be helpful.
2. In Table 2, the adaptive m setting does not outperform the fixed m version. Could the authors provide more insight into why adaptive m yields inferior results in this case?
3. The proposed test-time reasoning involves multiple sequential steps and the use of multiple vision-language models (VLMs). What is the overall inference speed?
4. The equations in the paper should be numbered for easier reference and clarity.

---

> ### Author Rebuttal · Authors · 2025-07-30
>
> $~~~~$We thank the reviewer for the thoughtful evaluation and for highlighting the paper’s advantages of (i) a unified treatment of detection + localisation + explanation and (ii) strong zero‑shot results without extra training. In the following parts of the rebuttal, we wish to address the concerns one by one:
>
> -----
>
> ### Major concerns
>
> > how sensitive is the method’s performance to the choice of hyperparameter l?
>
> $~~~~$We heuristically set our minimal suspicious window $\ell = \text{max}(300, T/10)$, in which 300 frames is a floor for the shortest window $W_{\text{max}}$. Since a clip $c_i$ (the smallest scoring unit) also spans $300~\text{frames} \approx 10s$, we expect that lowering this floor has little effect.
>
> $~~~~$As for the ceiling term $T/10$, we explored its stability by varying divisors on $T$.
>
> **Results:**
>
> |  | $\ell = \text{max}(300, T/5)$ | $\ell = \text{max}(300, T/10)$ | $\ell = \text{max}(300, T/15)$ |
> | --- | --- | --- | --- |
> | UCF AUC (%) | 81.07 | 84.28 | 83.66 |
>
> $~~~~$As a result, **an overly large $\ell$ (smaller divisor) degrades the performance**. We suspect that a large window size $\ell$ **hides fleeting anomalies** as the window may have higher probability of containing benign frames with lower scores, resulting in a lower estimate of the surrogate video-level anomaly probability $\tilde{s}_V$.
>
> $~~~~$In contrast, **smaller windows (larger divisor) remain stable** because the smaller window makes it **easier to find short-lived anomalies** with averaged scores near the natural decision boundary. We have included the ablation table above and discussion in the revised paper.
>
> $~~~~$In addition to the heuristics we used, it is also possible to introduce additional time-series segmentation model to identify events from sequences of frame scores.
>
> -----
>
> > Could the authors provide more insight into why adaptive m yields inferior results in this case?
>
> $~~~~$When setting a fixed $m$ value, we implicitly assumed **how ambiguiously the anomalies were defined** over the test samples.
>
> $~~~~$Empirically, a **smaller $m$ works well for UCF‑Crime and XD‑Violence where clearly defined crime/violent anomalies types may be obvious to the models**, so IntraTR can **"trust"** the first round prediction more by using a smaller $m$.
>
> $~~~~$However, on UBnormal, whose **anomalies are less clearly defined** (outlier human poses), the prior of **“smaller $m$” does not generalise well**, resulting in slightly lower performance. This result is expected, as greater ambiguity in anomaly definitions makes the initial estimation less reliable.
>
> $~~~~$Therefore, we offered an option of an adaptive $m$ estimating the per-video difference between normal and abnormal frames. It achieves consistently competitive performances across datasets, albeit sometimes slightly below the best fixed value. Thus, adaptive m serves as a reliable default bootstrap setting. We clarified such trade-off in **Table 7 and Section B1 (Lines 488-499)** of the submitted paper.
>
> -----
>
> > What is the inference speed of the full test-time pipeline involving multiple VLMs?
>
> $~~~~$To answer this question, we first analyse the complexity of our inference steps and compare it with the prior zero-shot VAD work, LAVAD (CVPR ’24).
> - Our test-time IntraTR pipeline for VAD requires 1 VLM captioning query and 1 LLM scoring query per 16 frames, along with a single VLM query per suspicious video to extract tags. For those videos flagged as “uncertain”, we perform an additional LLM scoring query per 16 frames. In total, our method performs at most 1 VLM and 2 LLM queries per 16 frames, plus fewer than 1 VLM query per video on average.
> - In contrast, full method of LAVAD (CVPR ’24) performs up to **5**  VLM captions **per frame** and **2** additional LLM queries for summarising and scoring per 16 frame. It also requires additional refinement steps that introduce computational costs of encoding massive captions and vector searching.
>
> $~~~~$The table below reports **amortised processing clock time inference speed** on 2 RTX 3090 GPUs for a full UCF‑Crime test set run (model loading time excluded):
>
> | Method | VLM Captioning (sec/frame) | Caption Cleaning (sec/frame) | LLM Summary (sec/frame) | LLM Scoring (sec/frame) | Score Refinement (sec/frame) | **VAD Overall (sec/frame)** |
> | --- | --- | --- | --- | --- | --- | --- |
> | LAVAD (CVPR 2024) | 0.06736 | 0.01490 | 0.01684 | 0.01109 | 0.00673 | 0.11692 |
> | **Ours** | **0.02587** | - | - | **0.00314** | **0.00026** | **0.02927** |
>
> $~~~~$Besides the VAD task, the VAL and VAU tasks each take standard inference time of the selected VLM (VAL: Qwen2.5-VL-7B, VAU: various models tested in the paper). The VAL takes one query per frame (by task formulation), while the VAU takes one query per video.
>
> $~~~~$As show by the results, despite neither method is real-time, LAVAD requires more LLM/VLM inference and takes longer time to refine the results compared with our method. We plan to include this discussion in further revision of the manuscript.
>
> -----
>
> ### Other weaknesses:
>
> > The equations in the paper should be numbered for easier reference and clarity.
>
> $~~~~$Thanks for pointing out, they have been numbered and referenced properly in the revised version.
>
> > The paper does not explicitly discuss the limitations or potential societal impacts of the proposed method.
> >
> $~~~~$We briefly discussed the **limitations and potential societal impacts** in **Section E, F (Lines 609-623)**. We have make it clearer by referencing to them in the main text in the revision.
>
> > Address possible failure cases, such as ambiguity in anomaly definition or limitations in zero-shot generalization to unseen domains.
>
> $~~~~$Thanks for the suggestion. We found some **failure cases when looking at normal or inconspicuous anomaly (e.g. Shoplifting) video samples**, where the model simply outputs either an empty tag $t_V$ list or neutral scene descriptions. e.g., for video visualised in **Fig. 7 (b),** the extracted $t_V = \text{``crosswalk, traffic light"}$.
>
> $~~~~$Since we are not allowed to post any external links or images during the rebuttal period, so these discussions and more qualitative analysis on failure have been included to the paper in the revision.
>
> -----
>
> $~~~~$We thank the reviewer again for the helpful feedback and high-quality review. We hope these clarifications address your concerns.

---

> ### Comment · Reviewer_Savy · 2025-08-04
>
> Thank you for the thoughtful rebuttal. I will be maintaining my current score.

---

> ### Author Response · Authors · 2025-08-05
>
> Thank you for your time and insightful review. Your valuable feedback has greatly helped us improve the quality of the manuscript. We have included the discussions and results in the updated version. We are happy to address any further limitations or concerns.

---

### Note · Authors · 2025-08-12

We thank all reviewers for their thoughtful feedback and the constructive discussion, which have helped us improve the paper. During rebuttal period, a majority of reviewers were willing to **raise their ratings** with the following addressed concerns:

Both **Reviewer Savy** and **Reviewer TbPh** raised concerns regarding **hyperparameter $\ell$ sensitivity** and **inference efficiency**. We provide additional ablations on $\ell$ demonstrating robustness, analyse the causes of degradation, and outline possible extensions to the original heuristic. We also clarified a specific hyperparameter concern from **Reviewer Savy** about the slight degradation when using adaptive $m$, by referring to the discussion in the main text.

For the concern on **inference efficiency**, we included a complexity analysis and wall-clock results showing that **our method is more efficient** than the prior baseline.

We addressed **Reviewer jCKt**’s concern about **missing zero-shot CoT baselines** by (i) running zero-shot experiments with a model (GLM-4.1V-9B-Thinking) **capable of long chain-of-thought (CoT) inference**, and (ii) referring to another **prompt-based reasoning approach (AnomalyRuler) evaluated in the main paper, which shows weaker zero-shot performance**.

We also provided **Reviewer jCKt** with a justification of our modular architecture compared with end-to-end multimodal models by adding further experiments and citing relevant findings from previous work.

We improved the presentation of the paper, addressing the ambiguities noted by **Reviewer DJMu**. We further emphasised (i) the **definitions of spatial metrics (Table 4, Lines 224–228)**, clarified (ii) the radius for **Gaussian smoothing** (set to the SciPy default), and (iii) the **methodological description** to avoid misunderstandings about frame sampling and model usage.

Additionally, we convinced **Reviewer TbPh** of the **novelty of our methodology**. Through existing and additional ablations (applying newer models to previous baseline → limited gains), we have shown that **the improvements arise from the design of the framework rather than from stronger foundation models**. We also adjusted the writing with (i) tightened component naming, (ii) referenced the necessary implementation details in the main text, and (iii) provided additional qualitative results.

We hope these changes lead to a stronger manuscript, and we thank all reviewers for their insightful and positive evaluation of this work.

---

### Decision · Program_Chairs · 2025-09-17

**Decision:**

Accept (poster)

**Comment:**

The paper introduces a unified reasoning framework for holistic zero‑shot video anomaly analysis, which leverages frozen vision–language models (VLMs) and large language models (LLMs) to perform, without any additional training, three core anomaly tasks: Temporal Video Anomaly Detection (VAD) via an Intra‑Task Reasoning (IntraTR) pipeline that computes frame‑wise anomaly scores, extracts contextual “anomaly tags”. Spatial Video Anomaly Localization (VAL) by chaining the VAD priors (tag list and refined scores) into a frozen VLM localization head, yielding bounding boxes for anomalous regions; Textual Video Anomaly Understanding (VAU)

Strengths:
+This paper designs a zero-shot approach for anomaly detection, which is more practical.
+Method validation across multiple anomaly detection frameworks demonstrates functionality on various modes of anomaly detection like VAU,VAD and VAL.
+Performance conducted on three benchmarks (CCTV, dash-cam/movie violence, synthetic scenes) show the high AUC over baseline approaches.

Weaknesses:
-The sub-task of spatial understanding lacks quantitative evaluation of spatial performance, while the qualitative visualizations are provided in the appendix.
-Some concerns about Gaussian smoothing including a sensitivity analysis regarding the choice of window size, should be addressed.
-According to the methodology, the authors use both VLMs and LLMs, but the framework diagram does not clearly indicate which specific model is used at each stage.


During rebuttal, some responses have been provided, such as, hyperparameter sensitivity and inference efficiency, more zero-shot CoT baselines, comparing with end-to-end multimodal models, etc.

After rebuttal, The final rating is 4 BA, and all reviewers agree the significant contribution of this paper.